# Effect of the Size and Shape of Dendronized Iron Oxide Nanoparticles Bearing a Targeting Ligand on MRI, Magnetic Hyperthermia, and Photothermia Properties—From Suspension to In Vitro Studies

**DOI:** 10.3390/pharmaceutics15041104

**Published:** 2023-03-30

**Authors:** Barbara Freis, Maria De Los Angeles Ramirez, Céline Kiefer, Sébastien Harlepp, Cristian Iacovita, Céline Henoumont, Christine Affolter-Zbaraszczuk, Florent Meyer, Damien Mertz, Anne Boos, Mariana Tasso, Sonia Furgiuele, Fabrice Journe, Sven Saussez, Sylvie Bégin-Colin, Sophie Laurent

**Affiliations:** 1UMR CNRS-UdS 7504, Institut de Physique et Chimie des Matériaux, CNRS, Université de Strasbourg, 23 Rue du Loess, BP 43, 67034 Strasbourg, France; 2Department of General, Organic and Biomedical Chemistry, NMR and Molecular Imaging Laboratory, UMONS, 19 Avenue Maistriau, 7000 Mons, Belgium; 3Tumor Biomechanics, INSERM UMR S1109, Institut d’Hématologie et d’Immunologie, 67091 Strasbourg, France; 4Department of Pharmaceutical Physics-Biophysics, Faculty of Pharmacy, Iuliu Hatieganu University of Medicine and Pharmacy, 6 Pasteur St., 400349 Cluj-Napoca, Romania; 5Inserm U1121, Centre de Recherche en Biomédecine de Strasbourg, 1 Rue Eugène Boeckel, CS 60026, CEDEX, 67084 Strasbourg, France; 6IPHC UMR 7178, CNRS, Université de Strasbourg, 67000 Strasbourg, France; 7Instituto de Investigaciones Fisicoquímicas Teóricas y Aplicadas (INIFTA), Departamento de Química, Facultad de Ciencias Exactas, Universidad Nacional de La Plata—CONICET, Diagonal 113 y 64, La Plata 1900, Argentina; 8Department of Human Anatomy and Experimental Oncology, Faculty of Medicine, Research Institute for Health Sciences and Technology, University of Mons (UMONS), Avenue du Champ de Mars, 8, 7000 Mons, Belgium

**Keywords:** iron oxide nanocubes and nanoplates, MRI contrast agent, magnetic hyperthermia, photothermia, targeting ligand

## Abstract

Functionalized iron oxide nanoparticles (IONPs) are increasingly being designed as a theranostic nanoplatform combining specific targeting, diagnosis by magnetic resonance imaging (MRI), and multimodal therapy by hyperthermia. The effect of the size and the shape of IONPs is of tremendous importance to develop theranostic nanoobjects displaying efficient MRI contrast agents and hyperthermia agent via the combination of magnetic hyperthermia (MH) and/or photothermia (PTT). Another key parameter is that the amount of accumulation of IONPs in cancerous cells is sufficiently high, which often requires the grafting of specific targeting ligands (TLs). Herein, IONPs with nanoplate and nanocube shapes, which are promising to combine magnetic hyperthermia (MH) and photothermia (PTT), were synthesized by the thermal decomposition method and coated with a designed dendron molecule to ensure their biocompatibility and colloidal stability in suspension. Then, the efficiency of these dendronized IONPs as contrast agents (CAs) for MRI and their ability to heat via MH or PTT were investigated. The 22 nm nanospheres and the 19 nm nanocubes presented the most promising theranostic properties (respectively, r_2_ = 416 s^−1^·mM^−1^, SAR_MH_ = 580 W·g^−1^, SAR_PTT_ = 800 W·g^−1^; and r_2_ = 407 s^−1^·mM^−1^, SAR_MH_ = 899 W·g^−1^, SAR_PTT_ = 300 W·g^−1^). MH experiments have proven that the heating power mainly originates from Brownian relaxation and that SAR values can remain high if IONPs are prealigned with a magnet. This raises hope that heating will maintain efficient even in a confined environment, such as in cells or in tumors. Preliminary in vitro MH and PTT experiments have shown the promising effect of the cubic shaped IONPs, even though the experiments should be repeated with an improved set-up. Finally, the grafting of a specific peptide (P22) as a TL for head and neck cancers (HNCs) has shown the positive impact of the TL to enhance IONP accumulation in cells.

## 1. Introduction

Regarding the synthesis of superparamagnetic iron oxide nanoparticles (SP-IONPs) for biomedical applications, most of the research is aimed at developing multifunctional theranostic nanoparticles that are able to either specifically target diseased organs or both identify disease states and deliver therapy, thus allowing the effect of therapy to be directly monitored by imaging. One of the current challenges is the design of SP-IONPs capable of specific targeting and combining both hyperthermia (via magnetothermia or photothermia) and magnetic resonance imaging (MRI) into a single nano-object with the highest possible efficiency to reduce the dose injected into the patient. The ability of SP-IONPs to be used as MRI T_2_ contrast agents (CAs) has already demonstrated [1,2], as well as their biodegradability and non-toxicity, compared to other CAs families [1,3]. Nevertheless, several issues have to be addressed to assure their use in medicine: a good colloidal stability in solution, provided by the anchoring of a ligand at their surface; a non-aggregated state; an optimal diffusion of water at their vicinity; a good biodistribution; and a high saturation magnetization (M_s_) [4]. As demonstrated in [5], there is still significant potential in developing nanoparticle-based MRI CAs to produce a more accurate, reliable, rapid, and targeted diagnosis of the suspected disease. 

Cancer treatment via magnetic hyperthermia (MH) is becoming a reality thanks to the positive results obtained by clinical trials conducted by MagforceTM (Germany) [6]. When exposed to alternating magnetic fields (AMF) of appropriate intensity (H) and frequency (f), SP-IONPs release heat locally (where they are concentrated) and can increase the temperature up to 40–43 °C, which reduces the viability of cancer cells, as they are known to be more sensitive to temperature compared to healthy cells. However, in most of the current in vivo strategies, MH therapy is combined with chemotherapy, radiotherapy, immunotherapy, photodynamic therapy, or gene therapy [7,8], as the SP-IONPs provide mild heating in the safety interval defined by H*f = 5 × 10^9^ Am^−1^s^−1^. Therefore, there is currently a need to improve the design of SP-IONPs for MH in order to reduce the dose injected into patients and to understand cell the death mechanism behind MH treatment. A particular strategy to make the MH treatment using SP-IONPs more efficient is to increase their M_s_ and effective magnetic anisotropy (K_eff_) [4,9,10]. The M_s_ can be increased by increasing the size of SP-IONPs close to the ferromagnetic state; thus, spherical IONPs of 20–25 nm in diameter are very promising candidates for combining both effective MRI and MH [11,12,13]. Another approach to increase MH as well as MRI efficiency is by adding the shape anisotropy to K_eff_, which can be achieved by tuning the SP-IONPs’ shape. Among the most interesting reported anisotropic SP-IONPs, nanocubes and nanooctopods stand out due to their excellent heating capabilities [13,14,15,16,17,18], as well as their interesting contrasting properties for MRI [19,20,21]. Apart from nanocubes, few papers focus on the synthesis of iron oxide nanoplates by the thermal decomposition method [18,21]. However, due to its strongly increased anisotropy, this shape could be of utmost interest for MH experiments. Previous experiments carried out by our team showed the potential of iron oxide nanoplates synthesized by thermal decomposition as MH thermal agents [17].

Finally, more recently, it has been demonstrated that SP-IONPs are able to induce a photothermal effect [22,23,24,25], and these results pave the way for a multimodal therapy. The mechanism behind IONPs’ photothermal effect is not entirely resolved but appears to be a result of d–d charge transfer between Fe^2+^ and Fe^3+^, which provides IONPs with a high adsorption of light in the near-infrared (NIR) region [23,24,25]. However, the effect of size or shape on PTT efficiency is not entirely clear. It has been shown that cubic NPs are more efficient than spherical counterparts [24,25], while the presence of a preferred lattice orientation in IONPs would favor heat release [26]. Other studies have suggested that neither the size nor the shape of IONPs plays an important role in efficient heating in the NIR [27]. Thus, it would be interesting to properly evaluate the effect of NP shape on the heat release by PTT; however, it appears that the IONPs’ crystallographic structure could be the key point to obtain an efficient heating agent under NIR irradiation.

Another important issue for biomedical applications is the coating of SP-IONPs by bioactive molecules, enabling their biocompatibility and good colloidal stability, and allowing the coupling of bioactive functions, such as dyes or targeting ligands [28]. Moreover, a major challenge in the theranostic approach is to be able to accumulate enough SP-IONPs in the tumor site while preventing the agglomeration of NPs in the physiological media. Indeed, many types of SP-IONPs have been developed for in vivo applications, but following intravenous injection, most of them accumulate in eliminatory organs, such as the liver, spleen, and kidneys, and only low amounts have been observed to accumulate in tumors [29,30,31,32]. In previous studies, we have developed a coating of SP-IONPs using polyethylene glycol (PEG)-based dendron molecules (dendronized IONPs: DNPs). Several in vitro and in vivo studies demonstrated that dendron coating ensures antifouling properties (no strong accumulation in the reticuloendothelial system (RES)) [13,17,33,34]. DNPs, which display a favorable biodistribution and bioelimination profile, can, thus, be used for affinity targeting in vivo and targeted therapeutic aims, due to their intrinsic hyperthermia properties. Previous DNPs coupled with melanin-targeting ligand (TL) showed a favorable biodistribution and biokinetic properties (no unspecific macrophage uptake and fast elimination in a few hours of the untargeted DNPs), and were homed to melanoma lesions and specifically internalized by melanoma cells [35]. Similarly, DNPs functionalized with a metronidazole-based ligand were demonstrated to be good candidates for the further development of agents recognizing hypoxic tissues, which is of high importance for both tumor investigation and in vitro tissue engineering [36]. Recently, a specific TL for the epidermal growth factor receptor (EGFR) was bound to the surface of 12 nm spherical DNPs by a carbodiimidation reaction to produce nanostructures able to specifically recognize EGFR-positive head and neck cancer cell lines (FaDu and 93-VU). The optimization of several reaction parameters ensured a high coupling yield and the quantification of the TL coupled to the DNPs, which was shown to be important to enhance their internalization, as compared to the bare nanoparticles [32]. The coupling of a TL at the surface of these DNPs represents a promising route to ensure and enhance their specific accumulation in tumoral cells.

In this study, iron oxide nanocubes and nanoplates were synthesized by the thermal decomposition method [13,14,17,37,38,39] and coated by dendron molecules. The MRI, MH, and PTT performances of the various colloidal suspensions were compared. Their magnetic properties were carefully studied and related to their properties. CAs for MRI and heating agents for MH and PTT effects were also evaluated in vitro. Finally, peptide 22, which coupled at the surface of DNPs, was optimized and grafted onto the IONP surface by a carbodiimidation procedure to determine the impact of the TL presence on IONP accumulation in cancerous cells.

## 2. Materials and Methods

### 2.1. Materials

Iron stearate (FeSt_3_) was either homemade from sodium stearate (purity 98.8%) or purchased from TCI (Zwijndrecht, Belgium), and ferric chloride was purchased from Sigma-Aldrich (Diegem, Belgium) or TCI. Iron stearate (FeSt2) was prepared from sodium stearate and ferrous chloride purchased from Acros Organics (Geel, Belgium). Organic solvents were purchased from Carlo Erba (Val-de-Reuil, France). Dioctyl ether (OE), 1-octadecene, HEPES buffer, N-hydroxy sulfosuccinimide (sulfo-NHS), sodium carbonate, and thiazolyl blue tetrazolium bromide (MTT) were purchased from Sigma-Aldrich. Oleic acid (OA) was purchased from Alfa Aesar (Haverhill, MA, USA). Dibenzylether (DBE) and squalane were purchased from Acros Organics. The dendron D1-2P was provided by Superbranche (Strasbourg, France). Peptide 22 (P22) was purchased from Polypeptide (Braine-l’Alleud, Belgium). Sodium hydrogen carbonate was purchased from Sigma-Aldricht (Diegem, Belgium). 1-Ethyl-3-(3′-dimethylaminopropyl) carbodiimide hydrochloride (EDC-HCl) was purchased from Carl Roth (Karlsruhe, Germany). Dimethyl sulfoxide was purchased from Fisher Scientific (Waltham, MA, USA). Reagents for cell culture: cell medium, PBS, Penicillin/Streptomycin 100X (P/S), trypsin-EDTA and fetal bovine serum (FBS) were purchased from Dominique Dutscher (Bernolsheim, France). The cell lines FaDu were donated by the Department of Human Anatomy and Experimental Oncology from the University of Mons (FaDu provided from ATCC (ATCC^®^ HTB43™) 

### 2.2. Method

#### 2.2.1. Synthesis of Iron Precursor FeSt_2_

Iron stearate (II) was prepared by the precipitation of sodium stearate NaSt and ferrous chloride FeCl_2_ salts in an aqueous solution [17]. 

#### 2.2.2. Synthesis of 10–12 nm Spherical IONPs

A 2.2 mmol FeSt3 (1.99 g) or FeSt2 (1.44 g) sample was added to 4.4 mmol of OA (1.24 g) in dioctylether (16.2 g, 20 mL). The mixture was heated to 120 °C for 60 min to dissolve the reagents in dioctylether. Then, a condenser was connected to the flask and the mixture was heated up to 290 °C with a 5 °C/min ramp. The mixture was refluxed at 290 °C for 120 min. The NP suspension was cooled down to 100 °C to proceed to the washing steps.

#### 2.2.3. Synthesis of 15–20 nm Cubic IONPs

To a 100 mL two-neck round-bottom-flask, 2.3 mmol of FeSt2 (1.44 g), 0.63 mmol of OA (0.18 g), and 2.29 mmol NaOl (0.70 g) (NaOl/OA ratio: 80/20) were added and mixed with containing octadecene (11.8 g, 15 mL). First, the mixture was heated to 110 °C for 60 min to dissolve the reagents in the solvent. Then, the condenser was connected and the mixture was heated up to 320 °C with a 5 °C/min ramp. The mixture was refluxed at 330 °C for 60 min. The black NP suspension was cooled down to 100 °C for the washing steps.

#### 2.2.4. Synthesis of Platelet-Shaped IONPs

In a 100 mL two-neck round-bottom-flask, 2.3 mmol of FeSt2 (1.44 g), 0.63 mmol of OA (0.18 g), and 2.29 mmol NaOl (0.70 g) (NaOl/OA ratio: 80/20) were mixed with octadecene (11.8 g, 15 mL). First, the mixture was heated to 110 °C for 60 min to dissolve the reagents in the solvent. Then, the condenser was connected and the mixture was heated up to 180 °C with a 5 °C/min ramp. The mixture was left at 180 °C for 60 min. The mixture was then heated up to 320 °C with a 1 °C/min ramp and refluxed at 320 °C for 60 min. The NP suspension was cooled down to 100 °C to proceed to the washing steps.

#### 2.2.5. Washing Step

The washing procedure is slightly changed, depending on the size and shape of the IONPs [17].

#### 2.2.6. Ligand Exchange (Dendronization) Step

Ligand exchange between OA and the dendron (D1-2P, Appendix A) was performed in THF according to previously published papers [34]. For nanoplates, nanocubes, and 20 nm spherical IONPs, the first amount of dendron added was raised to 10 mg and both mixing times were increased to 48 h. These changes were made to maximize the ligand exchange for larger NPs and faceted NP-like shapes.

#### 2.2.7. Targeting Ligand Grafting

The grafting was performed via a carbodiimidation reaction between the carboxylate groups of the dendron and the primary amines of the peptide [32].

#### 2.2.8. Characterization Techniques

**Transmission electron microscopy**. To investigate the size and morphology of the NPs, transmission electron microscopy (TEM) was performed with a JEOL 2013 microscope operating at 200 kV (point resolution 0.18 nm). 

**X-ray diffraction**. X-ray diffraction (XRD) patterns allow the crystalline phases of the as-synthesized IONPs to be determined. It is a quick way to identify core–shell nanoparticles with the characteristic peaks of the wüstite phase. However, it is more complicated to differentiate maghemite from magnetite as they display similar XRD patterns. Refinement of the XRD patterns is, thus, necessary to compare the lattice parameters of the IONPs to those of magnetite and maghemite phases and determine if the composition is closer to that of maghemite or magnetite. The diffraction patterns were refined by LeBail’s [40] method using Fullprof software [41]. The background, modeled as a linear function based on 20 experimental points, was refined, as well as the zero shift. The peaks were modeled with the modified Thompson–Cox–Hastings (TCH) pseudo-Voigt profile function.

**Fourier Transform Infrared Spectroscopy**. Standard infrared spectra were recorded between 4000 cm^−1^ and 400 cm^−1^ for both iron precursors using a Fourier transform infrared (FTIR) spectrometer Spectrum 100 from Perkin Elmer. The IONP samples were ground and diluted in a non-absorbent KBr matrix before analysis to assess the presence of the characteristic band between 800 cm^−1^ and 400 cm^−1^ corresponding to the spinel structure, the presence of the characteristic bands of oleic acid (alkyl bands between 3000 cm^−1^ and 2800 cm^−1^), and, more importantly, the disappearance of the characteristic band of the iron precursor at 720 cm^−1^.

**Dynamic Light Scattering**. The colloidal stability of the IONP suspensions in water (at 0.1 mg_Fe_/mL) and their mean hydrodynamic diameter (Dh) after ligand exchange were obtained by dynamic light scattering measurements performed on a Malvern Nano ZetaSizer instrument. The electrophoretic mobility was also recorded using the same equipment in order to obtain the zeta potential. It must be noted that for shaped IONPs, this measurement is an indication of possible aggregation but cannot be considered as a precise measurement of the hydrodynamic diameter, as the optical model used to calculate the size distribution considered spherical NPs.

**ICP-AES.** Inductively coupled plasma–atomic emission spectrometry analysis was performed to quantify iron and phosphorus contents within a colloidal suspension after an acidic attack with nitric acid [42]. When the ratio between the iron and phosphorous content, the surface, and the volume of a NP are known, it is possible to calculate the dendron grafting rate.

**Magnetic characterizations**. Magnetic measurements were performed with a superconducting quantum interference device (SQUID) magnetometer (MPMS SQUID VSM). To prepare samples, 50 µL of a suspension of NPs in chloroform or THF of known concentration was evaporated in the capsule (sample holder for SQUID apparatus). This step was repeated until a homogeneous layer of dried NPs covers the bottom of the capsule. Zero fields cooled/field cooled (ZFC/FC) curves were recorded as previously described [43].

**Nuclear magnetic resonance dispersion (NMRD) profiles**. NMRD profiles were registered on a fast-field-cycling relaxometer (Stelar, Italy), between 0.02 MHz and 40 MHz, at 37 °C. Additional T_1_ and T_2_ measurements were performed on a fixed field relaxometer (Minispec60, Bruker, Germany) working at 60 MHz and 37 °C.

**Relaxivity measurements**. T_1_ and T_2_ relaxation times of the DNPs suspensions were measured at 37 °C on a fixed-field relaxometer (Minispec60, Bruker, Germany) operating at a Larmor frequency of 60 MHz. Relaxivities r_1_ and r_2_ were obtained by measuring the longitudinal and transverse relaxation rate (R_1_ = 1/T_1_ and R_2_ = 1/T_2_) over an iron concentration range between 1.8 mM and 18 mM, by using the relations:R_i_ = 1/T_i_ − 1/T_water_ and r_i_ = R_i_/([Fe])

The relaxation time of the medium must be subtracted to obtain only the contribution of the DNPs. In our case, the medium is water and its relaxation times T_1_ and T_2_ are close to 3000 s. Since the relaxation time is very long, it does not impact the relaxation rates of the DNP suspension. Thus, it is possible to access r_1_, r_2_ (expressed in s^−1^·mM^−1^), and the ratio r_2_/r_1_.

#### 2.2.9. Magnetic Hyperthermia and SAR Evaluation in Colloidal Suspension

The heating performances of IONPs suspended in water or a solid matrix (PEG 8 k) were measured by a calorimetric method and quantified by the specific absorption rates (SAR). Two devices were used for MH experiments. The first device was the DM 100 instrument and DM applicator (Nanoscale Biomagnetics™, nB) with MaNIaC software. Vials adapted for magnetothermal measurements and filled with 1.0 mL of the samples at various iron concentrations ranging from 0.1 to 1.0 mg_Fe_/mL were submitted to AMF of −300 G (23.9 kA·m^−1^) amplitude and frequencies of 395 kHz and 536 kHz. The second device was EasyHeat 0224 from Ambrell (Scottsville, NY, USA), able to generate AMF with amplitudes from 5 to 65 kA·m^−1^ at a constant frequency of 355 kHz. The samples, held at 37 °C, consisted of 0.5 mL of IONP suspensions in water or PEG 8 k at an iron concentration of 1 mg_Fe_/mL. The increase in temperature was recorded for several minutes. The heating curves were plotted and their corresponding SAR values in W·g_Fe_^−1^ were calculated as described below. A second-order polynomial function was used to fit the heating curves during the first 60 s and to determine [dT/dt]_(t = 0)_. Finally, the SAR was calculated using the following equation:SAR = m_s_ × (C_s_/m_Fe_) × [dT/dt]_(t = 0)_
where m_s_ and C_s_ are, respectively, the mass (in g) and the heat capacity (in J g^−1^·K^−1^) of the sample; m_Fe_ (in g) is the mass of iron present in the sample; and [dT/dt]_(t = 0)_ the derivative function of the temperature at t = 0 (K·s^−1^). To determine this term, the heating curves were fitted with a second order polynomial equation whose equation is T(t) = T_0_ + [dT/dt]_(t = 0)_ × t − a × t^2^

Alternatively, the heating curves were fitted with the Box–Lucas equation:ΔT = (S_m_/k)(1 − exp(−k(t − t_0_)) 
where the fitting parameters S_m_ and k are the initial slope of the heating curve and the constant describing the cooling rate, respectively. The SAR can be calculated as
SAR = (c m S_m_)/m_Fe_
where c is the specific heat of colloid (water is 4186.8 J·kg^−1^·K^−1^ and PEG 8 k is 2135.27 J·kg^−1^·K^−1^); m = ρV is the mass of the colloid, taken as the product between the density (water is 0.997 g·cm^−3^ and PEG 8 k is 1.0832 g·cm^−3^) and the volume; and m_Fe_ is the mass of iron in the sample. IONPs, dispersed in water at the desired concentrations, were collected at the bottom of the vial by a magnet, the water was removed and hot liquid PEG 8 k (80 °C) was introduced. The dispersion of IONPs on the entire volume of the sample was ensured by immediate sonication with a probe sonicator, followed by sonification without/with the application of a static magnetic field of 65 kA·m^−1^ generated by two cubic neodymium magnets with a 3 cm side length.

#### 2.2.10. Photothermia Experiments and SAR Evaluation in Colloidal Suspension

A 2 mL sample of a known concentration of IONP suspension was placed in a 1 cm^3^ quartz cuvette (if the IONPs were suspended in organic solvent) or plastic cuvette (if the IONPs suspended in aqueous medium) and irradiated for 5 min with a near-infrared laser at 1064 nm and a power of 1 W·cm^−2^. A thermal probe was placed inside the cuvette to record the temperature during irradiation. After irradiation, the temperature increase as a function of time was plotted, and the associated specific absorption rate (SAR) in W·g_Fe_^−1^ was calculated from the experimental curve, as described below. By plotting the temperature profiles of the NP suspensions subjected to NIR light and then adjusting the experimental curve with a polynomial function, the [dT/dt]_(t = 0)_ can be determined. In order to perform precise calculations, the contribution of the solvent toward the temperature elevation must be removed. The calculation of the SAR is, thus, as follows:SAR = m_s_ × (C_s_/m_Fe_) × ([dT/dt]_(t = 0)_ − [dT/dt]_(t = 0, solvent)_)
where m_s_ and C_s_ are, respectively, the mass (in g) and the heat capacity (in J·g^−1^·K^−1^) of the sample; m_Fe_ (in g) is the mass of iron present in the sample; and [dT/dt]_(t = 0)_ is the derivative function of the temperature at t = 0 (K·s^−1^) of the sample and of the solvent only ([dT/dt]_(t = 0, solvent)_). Both of these derivatives are determined in the same way as for magnetic hyperthermia (i.e., a second-order polynomial fit of the temperature curve).

#### 2.2.11. Cell Lines Culture, Cell Viability Assay and Iron Uptake

These were performed as already described [32].

#### 2.2.12. Cell Observation with Transmission Electron Microscopy

In order to assess if the DNPs internalized well inside the FaDu cells, they were incubated with and without DNPs and imaged by transmission electron microscopy. The cells were first cultured in a 24-well plate with 5·10^4^ cells/well. After 24 h of adherence, the medium was changed and replaced with a new medium containing either no DNPs or DNPs at 100 µg_Fe_/mL. The cells were allowed to incubate for 24 h and were washed twice with PBS. Then, the cells were fixed in 2.5% glutaraldehyde in 0.05 M sodium cacodylate buffer at pH 7.4 for 30 min at room temperature and 2 h at 4 °C. The cells were then washed three times in 0.175 M sodium cacodylate buffer for 10 min. The samples were then post-fixed for 1.5 h in 1% osmium tetroxide in 0.15 M sodium cacodylate buffer. Samples were dehydrated by gradually increasing the ethanol concentration (30%, 50%, 70%, 95%, and 100%, 10 min every three times). Epoxy resin was prepared using the following: 48.2% Epon 812, 34% anhydride nadic methyl, 16.4% anhydride [2-dodecenyl] succinic anhydride, and 1.5% 2,4,6-tris dimethylaminoethyl phenol. Cells were transferred in a resin/100% ethanol mixture of 1:1 ratio for 30 min, then in of 2:1 ratio for 30 min, followed by 100% resin for 1 h two times. Finally, they were included in the fresh resin and polymerized for 48 h at 60 °C. After resin polymerization, heat shock was performed to remove the glass coverslips. Ultra-thin cross sections (100 nm) were obtained in a cultivation plan using an automatic ultramicrotome (Ultracut-E ultramicrotome, Reichert Jung, USA) and mounted on 100 mesh formvar carbon grids (Euromedex, Souffelweyersheim, France). Sections were stained with 5% uranyl acetate in 50% ethanol for 20 min and after rinsing with 4% lead citrate for 10 min, the grids were observed with a Hitachi H-7500 instrument (Hitachi High Technologie Corporation, Tokyo, Japan). The images were digitally recorded with an AMT Hamamatsu digital camera (Hamamatsu Photonics, Shizuoka, Japan).

#### 2.2.13. Magnetic Hyperthermia in Cells

Cells were seeded in an ibidi chamber (8 wells) with 104 cells/well and left for 24 h to undergo adherence. The medium was discarded, and 100 µL of new medium and 100 µgFe/mL of DNP suspension were used. The cells were allowed to incubate for 24 h with the new medium containing DNP. The medium was discarded and cells were thoroughly washed with PBS (3 times). Then, an alternating magnetic field of 305 kHz and 16 kA/m was applied to the cells for 1 h with the DM 3 instrument and DM applicator (Nanoscale Biomagnetics™, nB) using MaNIaC software, and after 24 h, cell viability was evaluated by an MTT assay or obtaining fluorescent images with a healthy/necrotic/apoptotic cell detection kit.

#### 2.2.14. Photothermia in Cells

Cells were seeded in a 96-well plate with 104 cells/well and left for 24 h to undergo adherence. The medium was discarded and replaced with 100 µL of new medium and 100 µgFe/mL of DNP suspension. The cells were left to incubate for 24 h with the new medium containing DNP. The medium was discarded and the cells were thoroughly washed with PBS (3 times). Then, the wells were irradiated for 5 min with a near-infrared laser beam set at 1064 nm and 1 W/cm^2^. After 24 h, cell viability was evaluated by performing either an MTT assay or obtaining fluorescent images with a healthy/necrotic/apoptotic cell detection kit.

## 3. Results and Discussion

### 3.1. Structural and Magnetic Characterizations of Various Shaped IONPs

To study the effect of the shape of IONPs on their performance as a contrast for MRI and heating agents by MH and PTT, nanocubes with a mean size of 14 and 18 nm and nanoplatelets (7 nm in thickness and 30 nm in length) were synthesized by the thermal decomposition (TD) method. Standard 12 nm sized nanospheres were also synthesized for comparison. TEM images of the different IONPs are given in Figure 1 and their mean size is reported in Table 1. The standard deviation for the TEM size is around 10% of the maximum for nanocubes and nanospheres (Table 1). The nanoplates sample is not quite homogeneous; it presents a second family of IONP shapes: smaller spherical NPs, visible on the TEM image in Figure 1.

FTIR spectroscopy and XRD analyses (Appendix A) show that all synthesized IONPs present a spinel structure with a composition close to that of magnetite Fe_3−X_O_4_. The IR spectra display the characteristic IR bands of iron oxide NPs coated with oleic acid and the Fe-O band is broad, with a maximum centered at 580 cm^−1^, which is characteristic of a slightly oxidized magnetite [44,45].

The refinement of XRD patterns allowed us to determine the lattice parameters and crystallite sizes, and they are reported in Table 1. By comparing the lattice parameters to those of bulk magnetite (8.396 Å, PDF file: 00-019-0629) and bulk maghemite (8.351 Å, PDF file: 00-039-1346), it is possible to assess how close the composition of the NP is to that of magnetite or maghemite. All samples display a composition close to that of magnetite. The average crystallite size can also be compared to the average TEM size. The observed discrepancies between both values could be linked to the presence of defects within IONPs, such as dislocations or antiphase boundaries [17,39,43,46,47,48]. Defects can arise from the TD process, which involves a wüstite nuclei Fe_1−X_O that grows and oxidizes to form the IONPs [47,49,50]. The oxidation process has been reported to generate defects [17,43,44]. An oxidized shell could also be responsible for such differences. Nevertheless, there are no strong differences between the crystallite and TEM sizes, suggesting a low amount of defects in our systems; this first observation is important as defects can have an impact on MH performance [4,43,48,51].

The magnetic properties of the different batches were determined by SQUID measurements. Magnetization curves were obtained at 5 K and 300 K and ZFC/FC curves under a magnetic field of 7.5 mT (75 Oe). Magnetization and ZFC/FC curves are reported in Figure 2. The saturation magnetization values at 300 and 5 K and the maximum of the ZFC curve assimilated to the blocking temperature are reported in Table 2. The saturation and remanent magnetizations were calculated for one gram of iron oxide Fe_3−X_O_4_.

The magnetization curves in Figure 2A are characteristic of a superparamagnetic behavior of all samples at 300 K. As expected, at 5 K, the magnetization curves (Figure 2B) display a hysteresis, indicating ferrimagnetic behavior at such a low temperature for all samples.

All selected samples exhibit a saturation magnetization above 50 emu/g at 300 K. The 12 nm nanospheres display a Ms value close to those already reported for such a size [44]. The nanoplates present a higher Ms than previously reported nanoplates (16.7 ± 5.2 (length), 5.7 ± 1.6 (thickness), and Ms = 43 ± 5 emu/g) [17,39], against the 20–25 nm length and the 7–8 nm thickness of our nanoplates batch), due to their higher length and thickness. Indeed, as both nanoplates are composed of oxidized magnetite (determined from the XRD study), we can assess that there is more magnetite in the larger nanoplates, thus exhibiting a higher Ms value. Nevertheless, this value remains quite low (as there are quite large NPs) and is attributed mostly to the proportion of smaller IONPs within the batch, contributing to a decrease in the M_S_ value, which is close to the value of the batch NP_12 at both 5 K and 300 K.

In contrast, the 19 nm cubes present a high M_S_ value at 5 K and 300 K, which makes them promising for magnetic hyperthermia and/or MRI, as they also present a high shape anisotropy. It is interesting to note that for the nanocubes, the size effect is quite visible when comparing M_S_ values of the 19 nm and the 14 nm cubes. The Ms increase could be explained by a larger fraction of magnetite, as Baaziz et al. also experienced [44]. Moreover, the Ms value could also increase with the NPs size due to a lower surface effect, such as the spin canting effect and fewer defects [28,44,52]. The nanocubes (14.5 nm ± 1.6 nm) of Cotin et al. [13] displayed a much smaller M_S_ value (39 ± 5) that the 14 nm nanocubes synthesized in this study because of their core–shell composition with a wüstite core. The nanocubes of Guardia et al. [53], with the same size (19 nm) as ours, also presented a homogeneous composition and the Ms values are similar (80 emu/g for Guardia et al. and 85 emu/g in this study). This could potentially enable high heating properties via MH for this nanocube batch.

Finally, the ZFC/FC curves are characteristic of superparamagnetic nanoparticles (Figure 2C). The Verwey transition around 100 K, characteristic of the magnetite phase, is observed in the ZFC/FC curves of NP_cubes_19, NP_cubes_14, and nanoplates (highlighted in yellow in Figure 2). Thus, the Ms values may be mainly explained by the presence of a higher amount of magnetite. The maximum of the ZFC/FC derivative curve is often assimilated to the IONP blocking temperature (Figure 2D). The blocking temperature values reported in Table 2 depend on the characteristics of nanoparticles and also on their aggregation state. Values are in agreement with reported values for nanospheres [44]. The values for nanocubes may be related to their higher size: a shift toward higher temperatures is observed for the nanocubes with the highest size. The blocking temperature value of nanoplates is close to that of spherical NP_12 but is difficult to discuss as this batch contains a large proportion of smaller spherical IONPs.

### 3.2. Ligand Exchange (Dendronization) Step

Ligand exchange was performed on all batches to exchange the oleic acid (OA) coating with the dendron molecule [33,34] (Appendix A). The ligand exchange has already been reported to be more difficult with faceted IONPs due to strong interactions in the oleic acid monolayer [13,17] and the ligand exchange time was extended. To maximize the ligand exchange on such faceted-shaped IONPs, such as the nanocubes and nanoplatelets, the dendronization step was performed twice, and the exchange time was increased to 48 h instead of 24 h which was usually used for 10/12 nm spherical IONPs. Briefly, after the first step of dendronization, the colloidal suspension was washed in an ultrafiltration unit to remove unbound OA and dendrons. Dendron molecules were then added again in excess and a second dendronization step was performed. The success of the ligand exchange step was verified by FTIR spectroscopy on the dendronized IONPs (DNPs). As shown in Figure 3, the characteristic bands of the dendron (C-O-C band around 1100 cm^−1^, phosphonate bands in the range 1100–800 cm^−1^, both highlighted in orange in Figure 3) are identified and the intensity of those characteristic of OA (alkyl band 3000–2800 cm^−1^, highlighted in green in Figure 3) decreases, confirming the replacement of OA by the dendron. The Fe-O bands for all batches are highlighted in yellow in Figure 3 and are compared to the theoretical IR spectrum of magnetite and maghemite, as shown in Appendix A. They confirm that the composition of all batches is in between that of magnetite and maghemite and corresponds to an oxidized magnetite Fe_3−X_O_4_.

ICP-AES analyses were performed on the dendronized batches to evaluate the grafting rate of the dendron molecule on the NPs surface (Table 3). The ICP-AES analysis is also a method to confirm the dendron presence on the NPs surface of each batch. Indeed, the analysis confirmed the presence of phosphorus in the four batches. The grafting rate of the batches DNP_12, DNP_cubes_14, and DNP_cubes_19 is similar: 1.4 dendron/nm^2^ for DNP_12 and DNP_cubes_14 and 1.5 dendron/nm^2^ for DNP_cubes_19. Unfortunately, it was not possible to properly evaluate the grafting rate for the plates batch. Indeed, in this batch, the nanoplates are also mixed with smaller spherical NPs, thus making it difficult to evaluate the mean surface of a NP.

The colloidal stability in deionized (DI) water was evaluated by measuring the hydrodynamic size distribution and the zeta potential values of each batch. The hydrodynamic size distribution is plotted in Figure 4 as the volume from DLS measurements and the mean hydrodynamic sizes are given in Table 4. The hydrodynamic size distribution is also plotted in intensity mode in Appendix A. All DNP suspensions display a good colloidal stability in water with quite a monomodal size distribution and with zeta potential values between −20 mV and −30 mV. Their mean hydrodynamic diameter is smaller than 50 nm for all batches. This is an important parameter to ensure that these DNPs will display a good biodistribution and pharmacokinetics properties during in vivo studies.

### 3.3. MRI Properties of the DNPs Suspensions

To assess the propension of DNPs to be good MRI contrast agents (CAs), their relaxation times T_1_ and T_2_ have been determined from relaxivity measurements. Relaxation time measurements were performed in the colloidal suspensions of nanoparticles in water at different iron concentrations ranging from 1.8 to 18 mM. From these data, relaxation rates R_1_ and R_2_ were plotted as a function of the concentration. The longitudinal and transverse relaxivity values r_1_ and r_2_ (expressed in s^−1^·mM^−1^) were calculated from the slope of the relaxation rates curves (Appendix A). A good T_2_ contrasting agent should exhibit a high relaxivity value of r_2_ and a high ratio of r_2_/r_1_. The longitudinal r_1_ and transverse r_2_ relaxivity values and their ratios are given in Table 4. A comparison was made with the commercial product Resovist and this commercial contrast agent is composed of small iron oxide cores of 3–5 nm clustered together. The mean hydrodynamic diameter of Resovist is around 60 nm.

The relaxivity r_2_ is plotted as a function of the saturation magnetization for the various batches at 300 K in Figure 5. Results coming from papers published by Cotin et al. [13,17] are also added to Figure 5 (gray shapes). For almost all batches (colored shapes), the higher the M_S_ value, the higher the ratio r_2_/r_1_. The relaxivity value r_2_ increases from 163 s^−1^·mM^−1^ to 407 s^−1^·mM^−1^ and the ratio increases from 20 to 29 for dendronized nanocubes of 14 and 19 nm, respectively. This is in agreement with the increase in the saturation magnetization with the IONPs size [4,54,55]. Interestingly, the cubes of Cotin et al. possess very high contrast agent properties, even though they possess a core–shell structure Fe_1−X_O@Fe_3−X_O_4_ and, thus, a lower magnetization saturation. Indeed, the relaxivity r_2_ is estimated to be at 332 s^−1^·mM^−1^ and r_2_/r_1_ = 30, which is equal to the results obtained with our cubic batch DNP_cubes_19.

The lower r_2_ and ratio values for the 12 nm sized DNPs are in agreement with their lower Ms values. However, nanoplates displaying a similar Ms to DNPs_12nm exhibit a higher r_2_ value and ratio. The nanoplates of Cotin et al. [17] present slightly smaller T_2_ contrast agent properties compared with our DNP_plates batch. Indeed, the r_2_/r_1_ ratio found for the nanoplates of Cotin et al. is about 15, whereas that of our nanoplates possess is 19; this difference can be linked to the smaller length and thickness of their batch, which also led to a smaller saturation magnetization. Indeed, the overall M_S_ is quite low (57 emu/g), whereas the r_2_/r_1_ ratio is high. This low M_S_ is attributed to the proportion of small spherical IONPs in the batch, whereas the high CA properties are certainly due to the nanoplates in the batch.

When compared to Resovist, it is important to note that almost all samples (except DNP_12) present higher relaxivity values r_2_ and a higher r_2_/r_1_, while having a smaller D_h_. The samples exhibiting the highest potential as T_2_ contrasting agents are DNP_cubes_19. Hence, the shape can also contribute to a more efficient CA.

Another possibility to evaluate DNPs’ efficiency as contrast agents is to record their proton nuclear magnetic resonance dispersion (NMRD) profile. This measurement corresponds to the relaxivity r_1_ evolution as a function of the frequency of the external magnetic field. In Appendix A, the NMRD profiles of the different batches exhibit a characteristic profile of superparamagnetic NPs with a bump around 1–3 MHz, whose position can be shifted due to the DNPs’ size (which is the case for batches DNP_plates and DNP_cubes_19). From the fitting of these NMRD profiles, the M_S_ values, as well as the NMRD radii, could be extracted, and are reported in Appendix A. The M_S_ values are always lower than the ones determined by SQUID measurements. Surprisingly, the value found for the cubes is always lower than expected (even lower than that for DNP_12). However, this can be explained by the fact that the fitting performed was not adapted for peculiar shapes, such as nanoplates or nanocubes, as the model is based on uniaxial symmetry.

### 3.4. MH Experiments

The heating performances of dendronized nanocubes and nanoplates were firstly evaluated in a water suspension as a function of iron concentration. The amplitude of AMF was fixed to 300 G (23.9 kA·m^−1^) while two frequencies were employed: 395 kHz and 536 kHz. The SAR values calculated for various iron concentrations (ranging from 0.1 to 1 mg_Fe_/mL) under both AMF frequencies are plotted in Figure 6, while their mean values are highlighted in Table 5.

As expected, the temperature increase is more significant when the frequency of the field increases. As an example, batches DNP_12 do not heat at 300 G and 395 K kHz whereas they are able to heat at 536 kHz. Dendronized nanocubes exhibited the highest heating values. This can be explained by their high shape anisotropy. Nanoplates (also exhibiting a higher shape anisotropy than nanospheres) do not present such high heating properties, which is certainly due to their lower Ms value. In addition, the MH heating is certainly hindered by the presence of smaller NPs within a batch.

To validate the interest of our batches as nano-heaters, their intrinsic loss powers (ILP, which corresponds to the heating capacity normalized by the strength and frequency of the applied magnetic field) were compared to other reported values in the literature presenting NPs of similar compositions and shapes, as shown in Table 6. DNP_cubes_19 and DNP_plates present high ILP values, higher than most of the reported values. Our batch of nanocubes presents particularly interesting heating properties, with an ILP estimated at 4.5 ± 0.7 nH·m^2^·kg^−1^. This value is not far from the outstanding heating capacity of the cubes synthesized by Guardia et al. (5.6 nH·m^2^·kg^−1^) and is higher than all the other nanocubes reported in Table 6. These better heating properties may be due to a chain arrangement of the cubes under the magnetic field, therefore increasing the anisotropy energy and, thus, the heating value, as observed in several publications [56,57,58,59,60]. To our knowledge, no iron oxide nanoplates comparable to ours have been reported in the literature and used for MH experiments. Our current nanoplates present both higher M_S_ values and a slightly larger length and thickness than those reported by Cotin et al. [13], so the shape anisotropy is also increased. Thus, the ILP value is also larger than those of previously reported plates.

In a second set of MH experiments, the heating performances of both nanocubes and nanoplates were investigated as a function of AMF amplitude (H), ranging from 5 to 65 kAm^−1^ at a particular iron concentration of 1 mg_Fe_/mL. As can be seen in Figure 7A, the SAR values of nanocubes increase linearly for H ranging from 5 to 30 kAm^−1^, and then saturates around 1780 W/g_Fe_ at the highest H values [63]. Please note that very high values of SAR at small H are within the safety limit (H*f = 5 × 10^9^ Am^−1^s^−1^) of MH treatment: 100 W/g_Fe_ for H of 5 kA/m, 300 W/g_Fe_ for H of 10 kA/m, and 570 W/g_Fe_ for H of 15 kA/m. At an H of 25 kA/m, the recorded SAR is 1050 W/g_Fe_, in accordance with the previous set of MH experiments (Figure 7A). As shown by the first set of MH experiments, the nanoplates display lower heating capabilities (Figure 7B). The SAR follows the same trend with increasing H, i.e., for H values up to 35 kA/m, the SAR increases linearly but with a lower slope as compared with the nanocubes (Figure 7). The saturation value of SAR is around 800 W/g_Fe_, which is less than half of the maximum SAR of nanocubes.

Overall, the nanocubes and nanoplates are promising for MH treatment thanks to their strong heating properties. However, one must not disregard the fact that the heating properties inside cells can be drastically diminished due to the confinement of lysosomes, which hinders the Brownian relaxation [64]. Thus, the heating performances of both types of IONPs were also studied in the solid matrix to determine how hampering the Brownian movement affects their heating capacities. As can be observed in Figure 7, the SAR value at each H decreases considerably when both types of IONPs are randomly immobilized in PEG 8 k. Compared with the water environment, nanocubes exhibit a mean drop in SAR values by 68%, while the nanoplates decrease by 78%.

This demonstrates that the heating is strongly affected by the highly viscous medium, and, thus, the heating may apparently be mainly governed by Brownian relaxation in water for these types of IONPs. This means that we expect lower heating performances in a confined environment, such as when IONPs are internalized in cells. In addition, it has been shown that anisotropic MNPs dispersed in water, under the dynamics induced by AMF, organize in chain-like structures [65,66]. Both the magnetization and anisotropy of these assembles are increased, leading to the enhancement of the SAR value through Neel relaxation rather than Brownian relaxation. The large difference between the SAR values recorded in water and PEG 8 K could be explained by invoking the above scenario in our case as well, although our NPs are coated with a dendron layer that might restrict the physical association of NPs with the formed chains.

Therefore, a possible way to make DNPs more efficient for in vitro and in vivo MH experiments is to increase the Neel relaxation contribution in the heating process. This implies the organization of DNPs into chain-like structures, when they are dispersed in the medium with reduced mobility [57,67]. In this regard, during solidification, the DNPs were exposed to a static magnetic field of 65 kA·m^−1^ strength (the maximum amplitude used in MH). The static magnetic field lines were parallel to those of AMF lines, meaning that the easy axes of magnetization of all DNPs were aligned parallel to each other and to the AMF lines. The arrangement of DNPS into chain-like structures might also be envisaged; however, the physical contact between DNPs is prohibited due to the dendron coating and solidified PEG 8 k. Despite this, neighbor DNPs might be close enough to manifest dipole–dipole interactions that develop a cooperative magnetic response to an AMF stimulus, as in the case of multicore flower-like IONPs [68,69]. A critical enhancement of SAR values within pre-aligned samples is, thus, expected. Indeed, as compared to a random case, the SAR values of pre-aligned nanocubes increased on average by 23% (Figure 7A). Surprisingly, the SAR values acquired in water are recovered by the pre-aligned DNPs for the first three values of H from 5 to 15 kA/m (Figure 7A). This is of colossal importance for cell and tumor MH treatments, since such applications are restricted by the imposed safety limit. By further increasing the H, the SAR increases slightly and saturates, forming a plateau already from a H of 30 kA/m (Figure 7A). The SAR values on the plateau are lowered on average by 45% with respect to those recorded in water. A closer inspection of Figure 7A reveals that the difference between SAR values recorded in water and in a pre-aligned configuration increases as the H is increased. These observations suggest that for a small H, the predominant heating mechanism is Neel relaxation, and starting with a H of 20 kA/m, Brownian relaxation comes into play and becomes increasingly relevant as H increases. In the case of nanoplates, the SAR values are enhanced on average by 20% (Figure 7B). For the first two H values, SAR values are close to those recorded in water, defining a plateau with a H of 40 kA/m. On average, the SAR drop is now by 58% with respect to that in water, suggesting that, for nanoplates, the Brownian mechanism is the relevant heating mechanism. Nevertheless, both types of IONPs are perfect candidates for in vitro and in vivo MH treatments as their pre-alignment retains the Neel magnetic relaxation in the solid matrix at small values of H [70].

### 3.5. PTT Measurements

The effect of the shape of IONPs on their performance as PTT agents was studied. The colloidal suspensions were irradiated for 3 min at various concentrations ranging from 0.1 to 1 mg_Fe_/mL with an NIR source of 1064 nm and a power density of 1 W/cm^2^. The corresponding SAR values are reported in Figure 8.

It must be noted that when the NP concentration increases from 0.1 to 1 mg·mL^−1^, the SAR values decrease quickly from 2500–3000 W/g_Fe_ to 500–1000 W/g_Fe_. A possible explanation is that the penetration depth of the laser in the sample is limited at high concentrations due to light absorption by the colloidal suspension [27,71]. This results in an inequal distribution of the incident light in the entire sample, with the DNPs in the front of the cuvette absorbing and scattering most of the light. This leads to a gradient of incident power in the sample with, on average, a lower absorption of the NPs [72]. This phenomenon was also observed by Perton et al. [73] and in other studies [27,71,72]. Nonetheless, the SAR value of our samples appears to be quite high at low concentrations. DNP-12 and DNP_cube_19 display a similar evolution in SAR values as a function of concentration. DNP_cube_19 present higher values at a high concentration compared to DNP_12, where DNP_12 display a high value at the lowest concentration. Surprisingly, DNP_plates display higher SAR values at low and intermediate concentrations: their SAR value at the lowest concentration is slightly higher than that of nanocubes. We cannot conclude unambiguously about an effect of the shape on these PTT SAR values from these data but they suggest insignificant effect of the shape (comparing DNP_12 and DNP_cubes_19) and an important influence of the IONP concentration on the PTT performance. Considering that concentration affects the laser beam diffusion, the SAR values at low concentration are suggested to be linked to the mean hydrodynamic size of the suspension. The DNP_12 suspension displays the lowest mean hydrodynamic size (21 nm) and the highest SAR value, which reaches 3000 W/g_Fe_ at a concentration of 0.1 mg_Fe_/mL; followed by DNP_plates with a mean hydrodynamic size of about 26 nm and a SAR value of 2547 W/g_Fe_ at a concentration of 0.1 mg_Fe_/mL; and finally DNP_cube_19, with a mean hydrodynamic size of 38 nm and a SAR value of 2453 W/g_Fe_ at a concentration of 0.1 mg_Fe_/mL.

### 3.6. Coupling the Targeting Ligand P22 and the Influence of P22 Presence on MRI Properties

The coupling of the peptide P22 (YHWYGYTPENVI), a peptide used as a targeting ligand for the EGF receptor [74] which is overexpressed in several cancerous cell lines such as head and neck cancerous cells [75,76,77], was performed by a carbodiimidation reaction [32]. The coupling was tested on the batches DNP_plate and DNP_cubes_14. To assess P22’s presence on the surface of the DNPs, the hydrodynamic diameter before and after the coupling reactions are compared (Table 7). A slight increase in the mean hydrodynamic size is observed, supporting the coupling of P22 on DNPs. However, some aggregation issues were noticed for DNP_plates+P22.

The relaxivity values of the batches before and after P22 grafting were compared to evaluate the impact of the TL on the MRI properties. Indeed, it is known from Pöselt et al. [78] that aggregation plays a key role in relaxivity values; thus, it is important to check if the relaxivity is impacted in batches that encountered aggregation issues after the coupling of P22. The results for DNP_12, DNP_cubes, and DNP_plates are reported in Table 7.

For DNP_12, P22 coupling at the IONPs surface does not seem to affect their relaxivity properties. Indeed, for both samples, r_2_ values remain quite high, and more importantly, the ratio between r_2_ and r_1_ is unchanged. For the cubes and plates, there is a slight decrease in the ratio r_2_/r_1_ after P22 grafting. As a reminder, the plate batch encountered slight aggregation after coupling; hence, a slight decrease in its MRI properties is not surprising.

### 3.7. In Vitro Studies

To explore the use of these DNPs as theranostic agents via MRI, MH, and PTT, their cytotoxicity and internalization in cells were studied. The cell lines studied are the head and neck cancer cell lines: FaDu cells, as they express the EGF receptor targeted by our chosen targeting moiety (P22) [79,80,81,82].

#### 3.7.1. Cytotoxicity Study

The cytotoxic effect of batches DNP_12, DNP_cubes_19, and DNP_plates was evaluated by an MTT assay after 24 h of incubation within the FaDu cells. The cell viability as a function of the iron concentration within the cell culture media for the four DNP batches is reported in Figure 9.

For the three batches DNP_12, DNP_cubes_19, and DNP_plates, the cell viability remains unchanged until 200 µgFe/mL and is almost not affected up to 400 µgFe/mL. Indeed, the cell viability stays higher than 80% for the three batches, at each concentration. Overall, our colloidal suspensions do not present any cytotoxicity up to a high level of concentration, which is very promising as it means that even if a high amount of DNPs are internalized within cells, they do not provoke a rise in cytotoxicity. This means that MH and PTT tests could also be performed at various concentrations. The viability study is in agreement with results previously reported by Cotin et al. [17], who studied the cell viability of Huh7 cells (liver cancer cells) incubated for 24 h with DNPs of various sizes and shapes (10 nm spherical DNPs, 22 nm spherical DNP, and 28 nm octopod DNPs) up to an even higher concentration. They proved that the DNPs of various shapes did not present any cytotoxicity up to an iron concentration of 4 mM (223 µgFe/mL) and even 8 mM (446 µgFe/mL) for some shapes. It is interesting to note from the internalization and cytotoxicity study that even if high amounts of DNPs are internalized within cells or localized at their surface, the cytotoxicity is not increased.

#### 3.7.2. Internalization Studies

DNP Batches without P22

A first test of internalization was performed on batches DNP_12, DNP_cubes, and DNP_plates. The batches were not conjugated to any targeting ligand here. The cells were incubated for 24 h with the above-mentioned colloidal suspension at an iron concentration of 100 µg/mL. The amount of iron per cell was evaluated by a Prussian blue colorimetric assay and the results are reported in Figure 10. As expected, DNP_12 did not internalize in cells at all, which is certainly due to their very high colloidal stability which cause them to not interact strongly with cells in plates. With the other batches, which display a lower colloidal stability and a higher mean hydrodynamic size, we were able to detect some DNP internalization. Indeed, DNP_plates display a lower mean hydrodynamic diameter in water and, thus, a lower cell internalization than DNP_cubes, which display the highest mean hydrodynamic size.

To confirm the above, DLS analyses were performed on batches of DNP_12, DNP_plates, and DNP_19_cubes in physiological media (i.e., a solution of 154 mM NaCl). The measurements were not performed in cell media as they contain many proteins that can disrupt the DLS measurement. The DLS results were obtained at 100 µg_Fe_/mL in water and in physiological media for the four batches, and are reported in Figure 11.

A major difference can be noted between the DLS size distribution in physiological media of batch DNP_12 and other batches. Indeed, the hydrodynamic diameter of DNP_12 in water or in physiological media stays the same. This batch is not disrupted at all by the presence of NaCl salt, whereas the other batch tends to partly aggregate in 154 mM NaCl solution. Thus, it is possible that the same phenomenon could occur in a culture cell medium, leading to a less stable suspension, with DNPs that tend to fall toward the bottom of the plate and are, thus, more easily internalized inside cells. This lower colloidal stability is not detrimental for in vitro studies. However, it is important to have a highly stable DNP suspension for in vivo study to ensure good pharmacokinetic properties of the DNPs after intravenous injection. Indeed, less stable suspensions could aggregate in vivo, which could, thus, be detrimental to patients.

To complete the study, transmission electron microscopy analysis was performed on the FaDu cell incubated for 24 h with either 100 µg/mL of DNP_cubes, DNP_plates, or without DNPs (control cells). TEM images of the three conditions are reported in Figure 12.

Both DNP types were found in the cells’ internalization lysosomes via an endocytosis transport pathway. Interestingly, many clusters of DNPs were also found on and next to the cell surface. These DNPs close to the cell surface were most certainly counted by the Prussian blue test presented above (Figure 11), so it is possible that the internalization results determined by the Prussian blue colorimetric assay of shaped and large-sized DNPs were overestimated due to these non-internalized aggregates. It must also be noted that many cells did not present any internalized DNPs. The shape of internalized DNPs is recognizable on the TEM images and it is interesting to note that the cubes tend to align in chains when they cluster together.

DNP batches with P22

After P22 coupling, the DNP_plate batch was very unstable. On the other hand, batches DNP_cubes_14 and DNP-12 remained globally stable after P22 coupling and it was proven that their relaxivity was almost unchanged after coupling. Therefore, the internalization of only these batches was investigated. The iron content was measured only in the FaDu cell line after 24 hrs and the content is reported in Figure 13.

In Figure 13, more internalization of DNPs within cells is observed when both DNPs (either the nanocubes or the nanospheres) are conjugated with P22. This result indicates that P22 may also increase internalization for larger-sized DNPs and for faceted DNPs, such as nanocubes. By comparing our results (with or without P22) with other internalization studies of IONPs within cells reported in the literature, it is found that the concentration for our DNPs is overall high. Indeed, it seems that the usual amount of internalized iron per cell after 24 h of internalization of IONPs at various iron concentrations is around 1 to 10 pg_Fe_/cell, depending on the size of DNPs and their aggregation state [83,84]. For example, Carenza et al. [85] showed an iron content of 7.2 pg of iron per cell for aggregated 7 nm IONPs coated with citrate after 24 h of incubation at 50 µg_Fe_/mL in murine outgrowth endothelial progenitor cells, whereas it was only 1.1 pg_Fe_/cell for the same dispersed IONP batch. Guerra et al. [86] observed between 4 and 6 pg_Fe_/cell for 6 nm IONPs incubated at 20, 50, and 100 µg/mL for 24 h in human glioblastoma cells. Other research groups demonstrated higher values (between 15 and 50 pg_Fe_/cell) when studying the internalization of clustered IONPs at various concentrations [87], which is more in the range of what we have shown in this study, even though it did not concern clusters of IONPs. The internalization results (mainly for DNP_22) may be overestimated due to the aggregation of DNPs within the culture cell medium. Indeed, some aggregates could be considered in the iron content determined during this test (not all aggregates were cleared by washes before the test).

#### 3.7.3. In Vitro Magnetic Hyperthermia and Photothermia Experiments

In vitro magnetic hyperthermia

From the colloidal suspension study, the batch DNP_cubes_19 presents a high potential for magnetic hyperthermia treatment. Moreover, internalization studies proved that they are partly internalized inside cells and partly present on the cell surface. Thus, in vitro MH experiments were carried out with this promising DNP suspension. To ensure no induction of cytotoxicity, the concentration in the cell culture medium was fixed at 100 µgFe/mL. This test was only performed on a monolayer of FaDu cells. They were incubated for 24 h either without DNPs or with DNPs at a concentration of 100 µg_Fe_/mL and washed thoroughly with PBS before MH treatment. The alternating magnetic field set at 304 kHz and 16 kA·m^−1^ (to respect the criterion H*f < 5·10^9^ A·m^−1^s^−1^ for clinical use) was applied for 1 h on cells. Afterwards, the cells were further incubated for 24 hours before labeling with a healthy/apoptotic/necrotic cell detection kit. The healthy cells are detected as a blue color by Hoechst DNA stain, the necrotic cells are detected by Ethidium homodimer in red, and the apoptotic cells are detected by Annexin V in green. Results are reported in Figure 14 with some fluorescent images of the two tested conditions (Figure 14A) and the percentage of healthy/apoptotic and necrotic cells for the two tested conditions is evaluated (Figure 14B).

The increase in the percentage of apoptotic and necrotic cells is slight but visible when the cells internalized with DNPs were exposed to AMF. The result of this preliminary test is promising but not entirely significant as the difference between the three cell percentages under the two conditions (with and without nanocubes) is quite low. This can be attributed to the fact that the main heating process of nanocubes is by Brownian relaxation, which is hampered when DNPs are internalized in cells. To more thoroughly evaluate the potential of this batch, a prealignment treatment could be performed, either in the same way as for the MH test in suspension in PEG 8k matrix, or by using an increased frequency and field, which were slightly above the criterion fixed for clinical use. Indeed, it was reported in the literature that even a slight increase above the H*f factor led to quite a different cell response with a higher cell death estimation [17]. Another interesting experiment would be to perform several cycles of AMF (in this way, it may be possible to stay below the H*f criterion for clinical use).

In vitro photothermia

Concerning PTT in vitro tests, contrary to MH, several samples were tested as they all seemed promising for PTT. As was the case for in vitro MH, only the FaDu cells were tested. They were incubated with 100 µgFe/mL of various DNPs: DNP_12, DNP_cubes_19, and DNP_plates) for 24 h in a 96-well plate. The wells were thoroughly washed with PBS and irradiated for 5 min with a near-infrared laser of 1064 nm at a power density of 1 W/cm^2^. Afterward, the cells were incubated for a further 24 h before measuring the cytotoxicity via the MTT assay. Unlike for MH, due to the high number of tested conditions, the effect of the combination of DNPs and laser irradiation was evaluated by an MTT cytotoxicity test rather than by taking fluorescent images of each condition. The results are reported in Figure 15.

It is interesting to note that the effect of the combination of DNPs and laser irradiation is visible in these experiments. Indeed, cell viability decreases when DNPs are combined with 5 min of irradiation for all DNPs. For example, the cell viability is estimated as 81% for DNP_cubes without laser irradiation and drops to 68% after laser irradiation. The same effect is observed for DNP_12 (97% cell viability without laser and 78% with the addition of laser) and DNP_plates (100% cell viability with DNP_plates only and 85% when a laser is added). The laser only (not combined with a DNP suspension) seems to have less impact on the cell viability (93% of cell viability with only laser irradiation). These results are quite promising for the therapeutic effect of our DNPs as a photothermal agent. Moreover, several irradiation cycles could be performed to determine if the temperature increase effect is improved with the number of cycles. The power settings or irradiation time could also be changed to determine how these parameters affect cell death. In the literature to date, it appears that in vitro PTT experiments have been carried out for no more than 10 min and with a power density ranging from 0.3 W/cm^2^ to 1 W/cm^2^ [24,71,88,89,90]. Thus, irradiation time, the impact of cycles, and laser power are parameters to be explored in more depth in future studies.

## 4. Conclusions

The MRI, MH, and PTT properties of DNPs of various sizes and shapes were tested and the main conclusions are summarized in Table 8. Increasing the size and shape anisotropy proved to be an efficient way to increase the relaxivity r_2_ values and the ratio r_2_/r_1_ of DNPs. Thus, large-sized DNPs and anisotropic DNPs, such as DNP_cubes_19 or DNP_plates, are promising to be used as a contrast agent for MRI. Moreover, we have proven that even after the coupling of P22, the MRI properties are unchanged as long as the colloidal stability of the suspension remains high. Concerning magnetic hyperthermia, we have demonstrated that the 19 nm cubic batch (DNP_cubes_19) presents an exceptionally high heating performance thanks to its high saturation magnetization value. The study in an immobilized matrix led us to conclude that the heating is mainly due to Brownian relaxation; however, the heating can be increased and even recover the values found in water medium for low fields if the DNPs are realigned in the matrix. Another interesting aspect consider for future studies is the impact of defects on relaxation movements, according to the studies of Lak et al. [48]. DNP_cubes_19 led to very high SAR and ILP values compared to reported values, and, hence, its selection for in vitro MH studies. Finally, for PTT, all sizes and shapes heat intensely when irradiated by near-infrared light. It was difficult to conclude the effects of shape and size for this study. Thus, an in vitro PTT assay with batches of various sizes and shapes was carried out. Thinking in terms of a dual treatment to take advantage of PTT and MH and reduce the NP dose injected [24], it appears that high-length cubes (presenting a Fe_3−X_O_4_ composition), such as batches DNP_cubes_19, would be well suited. Indeed, they heat well under laser irradiation and have a strong potential for MH treatment. Moreover, some studies suggest that laser irradiation could regenerate the Brownian motion, which was lost due to immobilization inside cells [62]. Thus, in the future, it would be interesting to explore their use for dual treatment in depth by studying their therapeutic application in vitro.

In vitro studies allowed us to observe that our large-sized and shaped DNP batches internalize in quite high amounts in cells. Even though this is partly due to a higher aggregation phenomenon in physiological media, which may slightly overestimate the iron content found in cells, the TEM study proved that DNPs are accumulated inside cells, and also partly on their surface. The cytotoxicity revealed that there is no strong cytotoxicity up to quite a high concentration (200 µg_Fe_/mL) for almost all of the tested batches. The strong internalization of DNPs into cell lysosomes and on their surface does not seem to impact their viability, which is really promising for further therapeutic tests.

We were able to perform several preliminary in vitro MH and PTT tests, which confirmed the promising effect of our DNP batches as nano-heaters via MH and PTT. However, due to set-up issues that should be resolved for future tests, at present, we could not accurately undertake repeatability studies of in vitro MH and PTT. A further interesting study for MH or PTT would be to monitor the temperature increase during MH and PTT experiments and to perform several cycles of either AMF or laser irradiation to determine check the effect of cycles on cell viability. Through the studies performed in suspension and in vitro, we suggest that our DNP batches, particularly the batch DNP_cubes_19, could be well suited for a dual treatment combining MH and PTT.

## Figures and Tables

**Figure 1 pharmaceutics-15-01104-f001:**
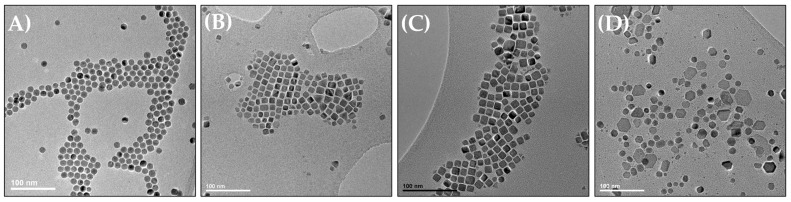
TEM images of (**A**) 12 nm sized nanospheres, (**B**) 14 nm sized nanocubes, (**C**) 19 m sized nanocubes, and (**D**) 30 nm length and 7 nm thickness nanoplatelets.

**Figure 2 pharmaceutics-15-01104-f002:**
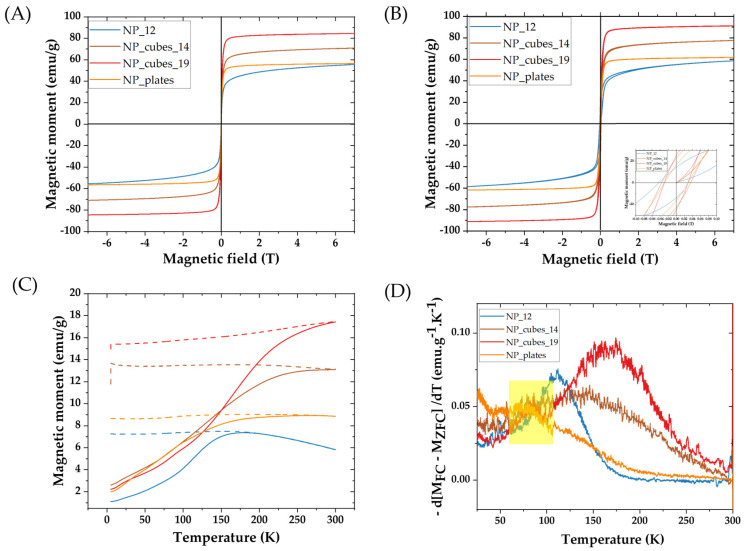
Magnetization curves at (**A**) 300 K and (**B**) at 5 K (inset: zoom around 0 T) of different IONPs. (**C**) ZFC/FC curves under a magnetic field of 7.5 mT (75 Oe) and (**D**) derivative of ZFC/FC curves, with the Verwey transition highlighted in yellow.

**Figure 3 pharmaceutics-15-01104-f003:**
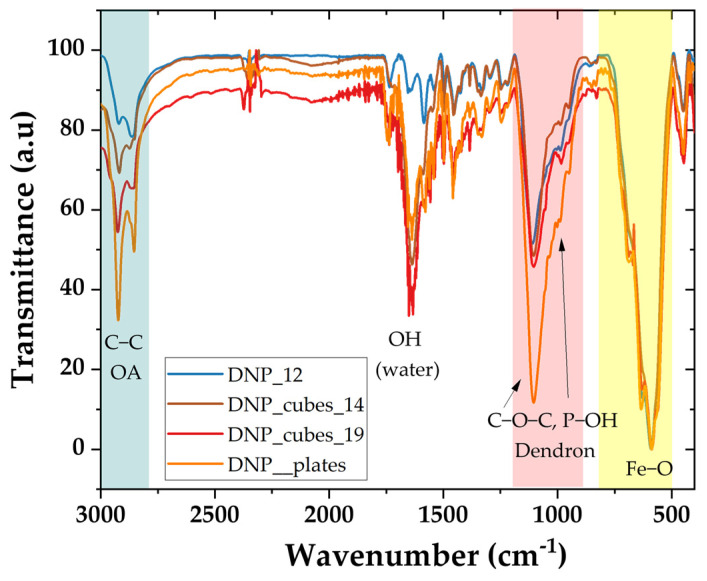
FTIR spectra of dendronized selected batches, with the characteristic bands of the Dendron highlighted in red; the alkyl bands of OA, which are less intense, highlighted in green; and the Fe-O IR bands highlighted in yellow.

**Figure 4 pharmaceutics-15-01104-f004:**
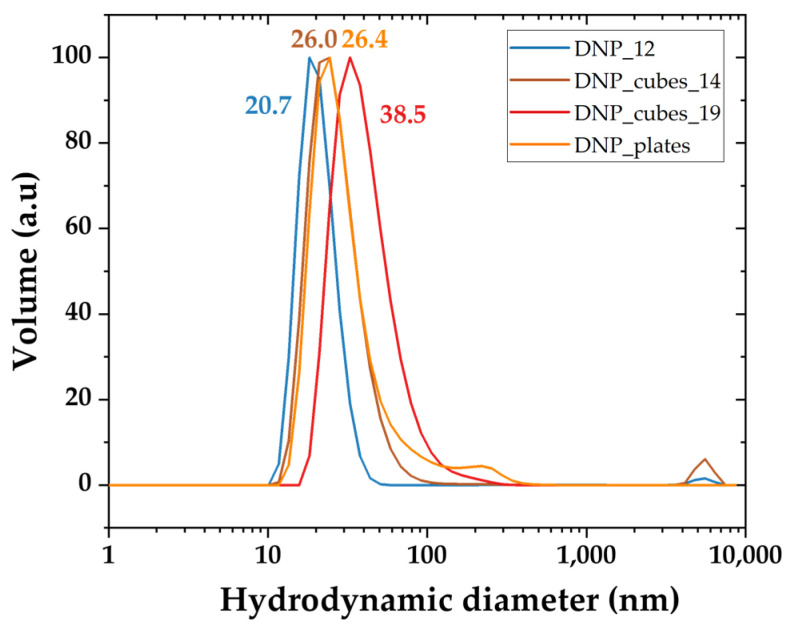
Hydrodynamic size distribution in volume of all DNPs measured by DLS.

**Figure 5 pharmaceutics-15-01104-f005:**
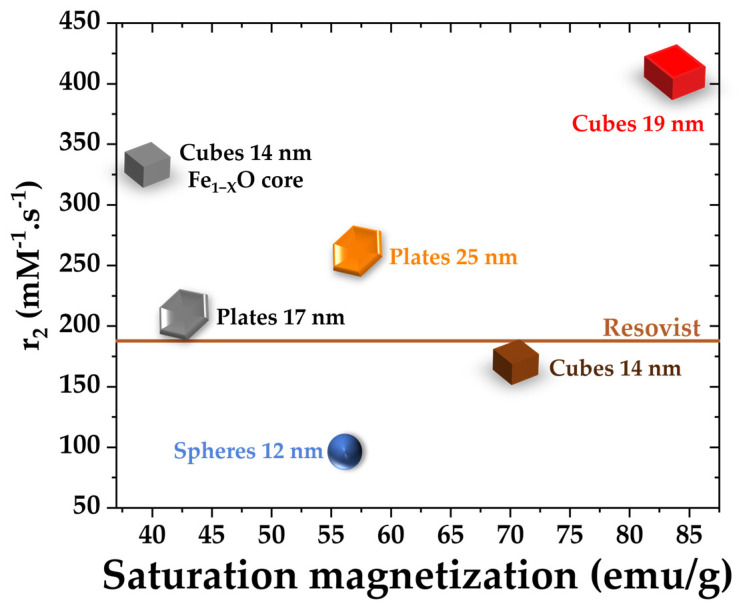
Relaxivity value r_2_ as a function of the saturation magnetization of the various batches. Results in gray are from Cotin et al. [13,17].

**Figure 6 pharmaceutics-15-01104-f006:**
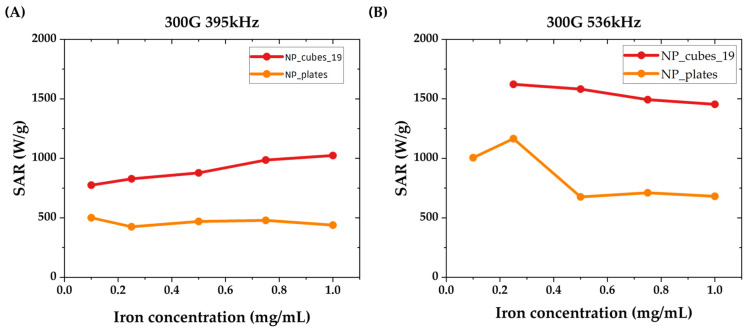
SAR values measured at various Fe_3_-xO_4_ concentrations at (**A**) 300 G, 395 kHz and (**B**) 300 G, 536 kHz.

**Figure 7 pharmaceutics-15-01104-f007:**
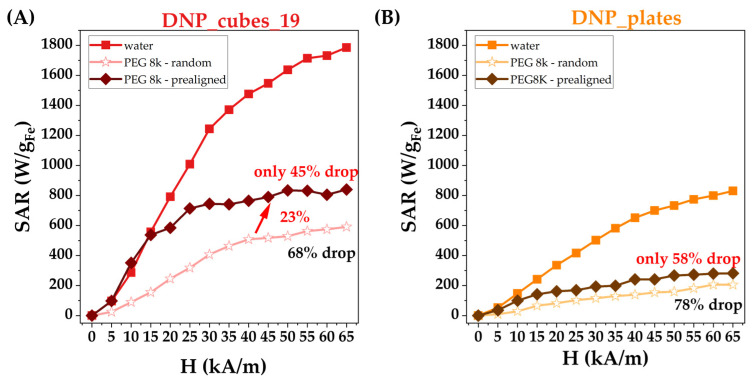
SAR values as a function of AMF in water and in the PEG 8 k matrix with and without prealignment using a magnet for (**A**) the DNP_cubes_19 suspension and (**B**) the DNP_plates suspension measured at 1 mg_Fe_/mL.

**Figure 8 pharmaceutics-15-01104-f008:**
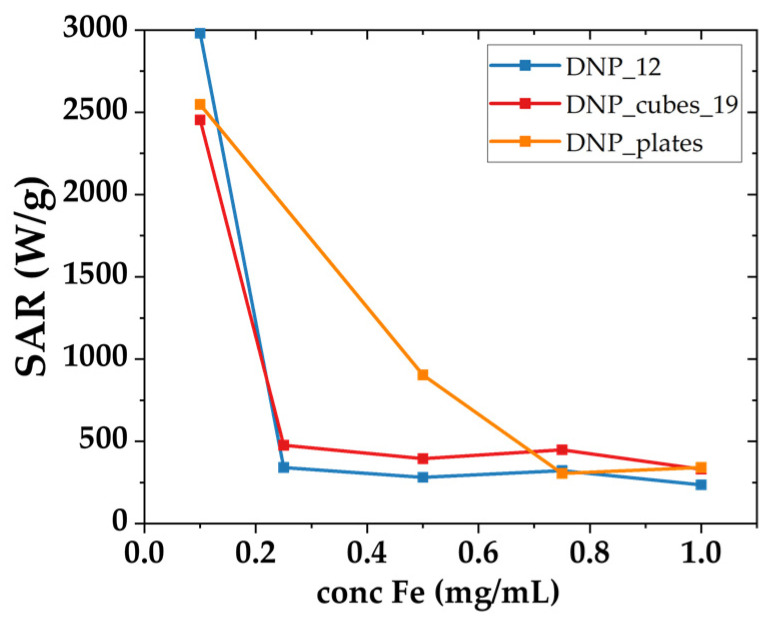
SAR values as a function of iron concentration in mg/mL for the batches DNP_12, DNP_cubes_19, and DNP_plates.

**Figure 9 pharmaceutics-15-01104-f009:**
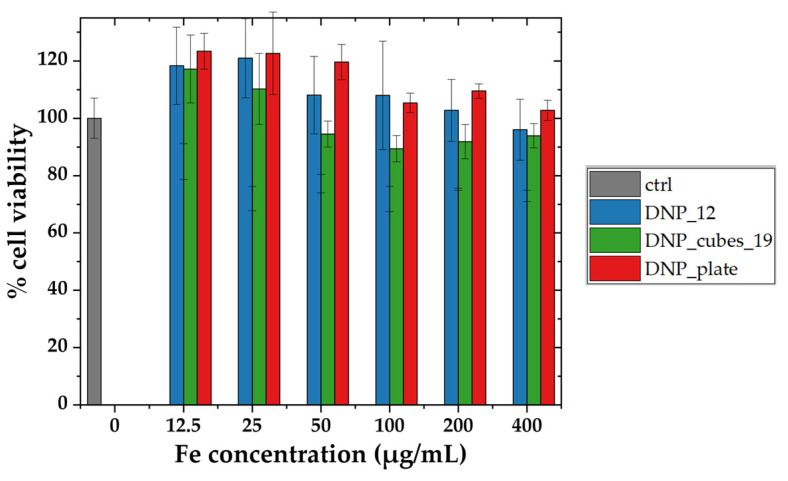
Cell viability of FaDu cells measured by the MTT assay as a function of iron concentration in µg/mL for DNP_12, DNP_cubes_19, and DNP_plates after 24 h of incubation with FaDu cells.

**Figure 10 pharmaceutics-15-01104-f010:**
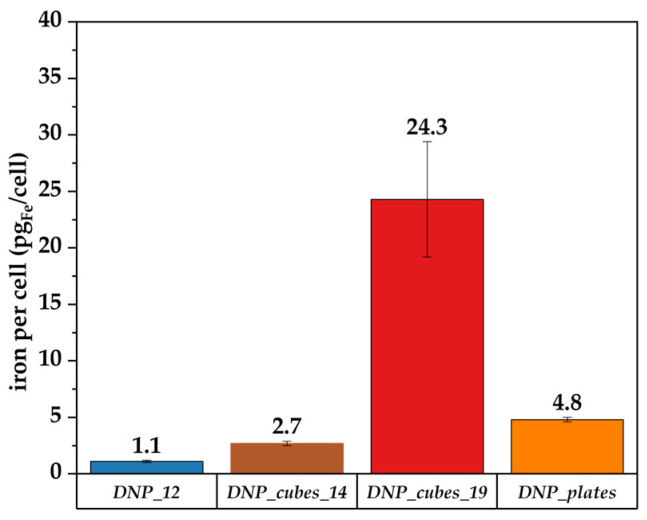
Iron content in FaDu determined by Perls’ Prussian blue colorimetric method after their incubation with DNP_12, DNP_cubes_19, and DNP_plates for 24 h and at an iron concentration of 100 µg/mL.

**Figure 11 pharmaceutics-15-01104-f011:**
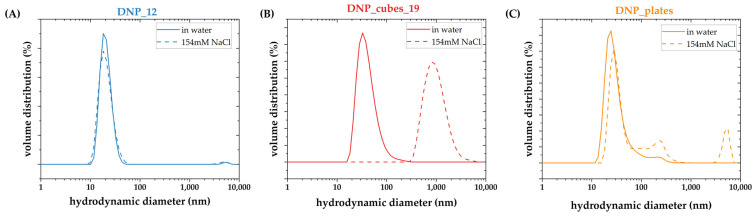
DLS size distribution in the volume of (**A**) DNP_12, (**B**) DNP_cubes_19, and (**C**) DNP_plates measured at 0.1 mg Fe/mL in water (straight line) or in physiological media (154 mM NaCl solution) (dashed line).

**Figure 12 pharmaceutics-15-01104-f012:**
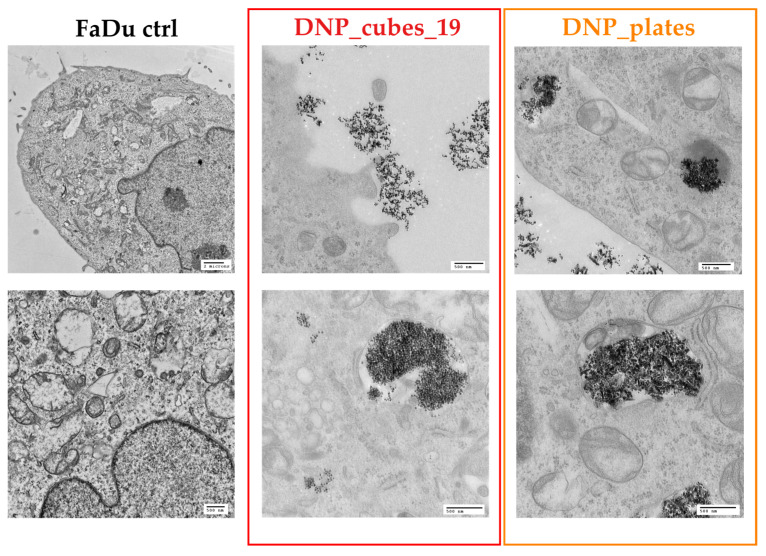
TEM images of FaDu control cells (left images), FaDu cells incubated with DNP_cubes_19 for 24 h at 100 µg/mL (images in the red square), and FaDu cells incubated with DNP_plates for 24 h at 100 µg/mL (images in the orange square).

**Figure 13 pharmaceutics-15-01104-f013:**
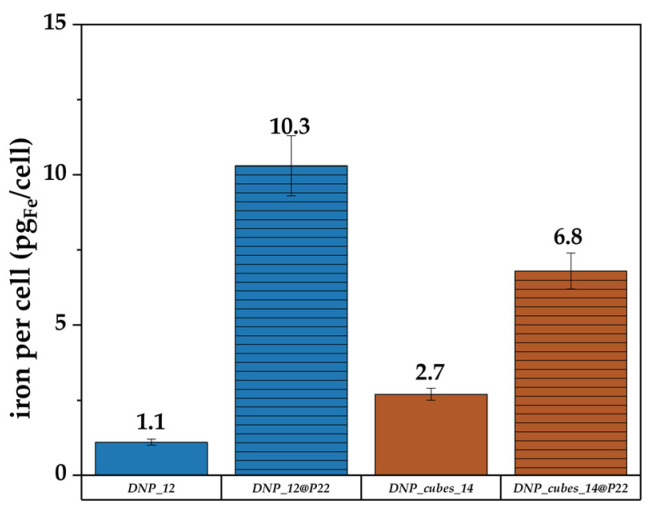
Iron content in FaDu cells determined by the Perls’ Prussian blue colorimetric method after their incubation with DNP_12, DNP_12+P22, DNP_cubes_14, and DNP_cubes_14+P22 for 24 h and at an iron concentration of 100 µg/mL (lined bars = DNP + P22).

**Figure 14 pharmaceutics-15-01104-f014:**
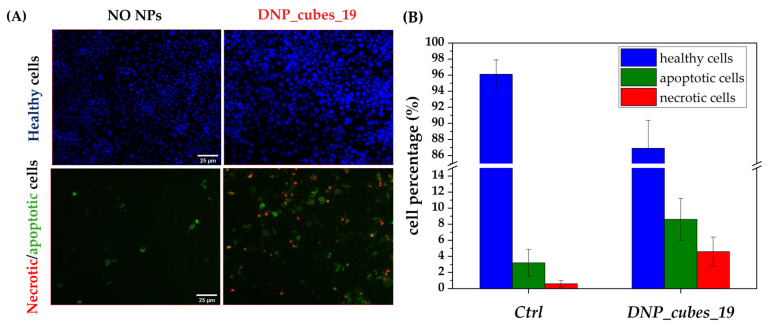
(**A**) Fluorescent images of FaDu cells 24 h after exposure to AMF (304 kHz, 16 kA·m^−1^) with and without DNP_cubes_19 (incubated for 24 h at 100 µg/mL). (**B**) Corresponding percentage of healthy, apoptotic, and necrotic cells for both cell conditions.

**Figure 15 pharmaceutics-15-01104-f015:**
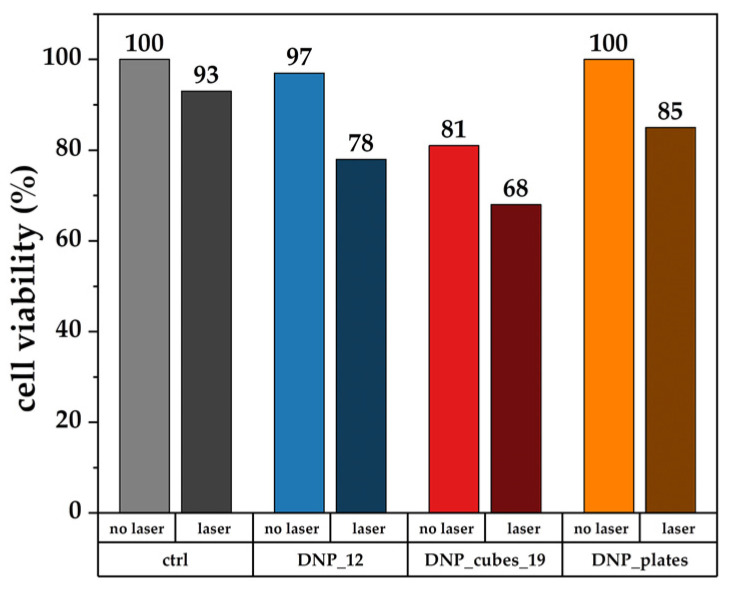
FaDu cell viability measured by the MTT test 24 h after 5 min of irradiation with a 1064 nm laser of 1 W/cm^2^. Cells were either incubated with no DNPs (control, ctrl) or with 100 µg/mL of DNP_12, DNP_cubes_19, and DNP_plates.

**Table 1 pharmaceutics-15-01104-t001:** Average TEM size, crystallite size, and lattice parameter of various batches.

Batch	TEM Size (nm)	Crystallite Size (nm)	Lattice Parameter (Å)
12 nm nanospheres	12.8 *±* 1.1	8.0	8.388
14 nm nanocubes	13.7 *±* 1.3	11.2	8.392
19 nm nanocubes	19.6 *±* 2.0	17.8	8.374
Nanoplates	Length: 25–30 nm Thickness: 7–8 nm	15.0	8.382

**Table 2 pharmaceutics-15-01104-t002:** Saturation magnetizations (M_S_, measured at 300 K and 5 K) and the maximum of the ZFC curve (T_max ZFC_) assimilated as the blocking temperature of the various batches.

Batch	M_S_ (emu/g) 300 K	M_S_ (emu/g) 5 K	T_max ZFC_ (°C)
12 nm nanospheres	56	59	112
14 nm nanocubes	71	78	145
19 nm nanocubes	85	91	170
Nanoplates	57	62	120

**Table 3 pharmaceutics-15-01104-t003:** Fe and P content evaluated by ICP-AES analysis and the grafting rate of the dendron in the various batches.

Batch	P Content	Fe Content	Grafting Rate (Dendron/nm^2^)
DNP_12	954 ± 9	17 ± 1	1.4
DNP_cubes_14	492 ± 7	8.2 ± 0.1	1.4
DNP_cubes_19	1122 ± 15	14 ± 2	1.5
DNP_plates	889 ± 11	30.4 ± 0.2	/

**Table 4 pharmaceutics-15-01104-t004:** Mean hydrodynamic diameter and relaxivity values of selected batches of DNPs measured at 60 MHz and 1.41 T, compared with Resovist, a commercial T_2_ contrast agent for MRI.

Batch	D_h_ (nm)	r_1_ (s^−1^·mM^−1^)	r_2_ (s^−1^·mM^−1^)	r_2_/r_1_
DNP_12	20.7 ± 0.2	12	95	8
DNP_cubes_14	26.0 ± 0.4	8	163	20
DNP_cubes_19	38.5 ± 0.6	14	407	29
DNP_plates	26.4 ± 0.4	13	250	19
Resovist	60	9	189	19

**Table 5 pharmaceutics-15-01104-t005:** Mean SAR value of selected batches at two different AMFs.

Batch	Field Amplitude (kA·m^−1^)	Field Frequency (kHz)	SAR (W/g)
DNP_cubes_19	23.9	395	899 ± 105
		536	1538 ± 78

DNP_plates		395	463 ± 31
		536	847 ± 224

**Table 6 pharmaceutics-15-01104-t006:** Comparison of ILP values of DNP_cubes_19 and DNP_plates with reported values for similar shaped IONPs.

Batch or Reference	Composition	Shape	Core Size (nm)	Coating	Solvent	M_S_ (emu·g^−1^) at 300 K	ILP (nH·m^2^·kg^−1^)
**DNP_cubes_19**	**Fe_3−X_O_4_**	**cubes**	**19.6 ± 2.0**	**dendron**	**H_2_O**	**85**	**4.5 ± 0.7**
[56]	Fe_3−X_O_4_	cubes	~16	dendron	H_2_O	/	1.1
[13]	Fe_1−X_O@Fe_3−X_O_4_	cubes	14.5 ± 1.6	dendron	H_2_O	39	1.5
[53]	Fe_3−X_O_4_	cubes	~19	PEG	H_2_O	80	5,6
[61]	Fe_3−X_O_4_	cubes	20	oleic acid	DMSO/H_2_O	89	1.1
[24]	Fe_3−X_O_4_	cubes	20	PEG	H_2_O	/	2.2
[62]	Fe_3−X_O_4_	cubes	18	PEG	H_2_O	/	2.9
[48]	Fe_3−X_O_4_	cubes	16	PEG	H_2_O	55	2.2
**DNP_plates**	**Fe_3−X_O_4_**	**plates**	**Length: 25–30** **Thickness 7–8**	**dendron**	**H_2_O**	**57**	**2.5 ± 0.5**
[13]	Fe_3−X_O_4_	plates	Length: 16.7 ± 5.2 Thickness 5.7 ± 1.6	dendron	H_2_O	57	2.1

**Table 7 pharmaceutics-15-01104-t007:** Relaxivity values and mean hydrodynamic diameter of various batches of DNPs before and after P22 grafting measured at 60 MHz and 1.41 T.

	DNPs				DNPs + P22			
Batch	r1 (s^−1^·mM^−1^)	r2 (s^−1^·mM^−1^)	r2/r1	D_h_ (nm)	r1 (s^−1^·mM^−1^)	r2 (s^−1^·mM^−1^)	r2/r1	D_h_ (nm)
**DNP_12**	10	86	9	20.7	14	130	9	21.7
**DNP_cubes_14**	8	163	20	23.0	10	181	18	26.1
**DNP_plates**	13	250	19	26.4	21	354	17	26 + peak at 2000 nm

**Table 8 pharmaceutics-15-01104-t008:** Main conclusions on the effects of size, shape, and the presence of a TL on MRI, MH, and PTT.

	Main Conclusions	Best Batches
**MRI**	Increase in shape anisotropy and size No change after P22 coupling	DNP_cubes_19, DNP_plates
**MH**	High saturation magnetization Heating mainly comes from Brownian relaxation	DNP_cubes_19
**PTT**	Hard to conclude on the effect of size or shape	All batches
**In vitro studies**	Strong internalization for larger-sized and shaped DNPs due to an aggregation phenomenon in cell media Cell viability not affected up to 200 µg_Fe_/mL	/

## Data Availability

The data that support the findings of this study are available in the Appendix A of this article. The data are also available on request from the corresponding author. The data are not publicly available because the authors would like to present them at conferences.

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
