# Peer review of "Effect of the Size and Shape of Dendronized Iron Oxide Nanoparticles Bearing a Targeting Ligand on MRI, Magnetic Hyperthermia, and Photothermia Properties—From Suspension to In Vitro Studies"

_pharmaceutics, 2023, doi:10.3390/pharmaceutics15041104_

Round 1

Reviewer 1 Report

The author presented the paper "Effect of the size and shape of dendronized iron oxide nanoparticles bearing a targeting ligand on MRI, magnetic hyperthermia, and photothermia properties - from suspension to in vitro studies"

It is an interesting  work. However, below are some questions that can improve the quality of the article.

1) The reference list should be improved. Many more 2-3 years review papers should be cited in the Introduction section to show the progress in the area. I highly recommend not to use the references older 10 years for this section if it is possible.

2) The first thing that is important for treatment and MRI is solubility in water. Please, clearly mention for the resulted system their colloidal stability, solubility in water. However, I have read done the text that they form an emulsion. It is not good for biomedical application.

Please mention the solvent and concentration of nanoparticles and their form in this solution (good solubility, emulsion, etc.) for all methods in experimental sections and figure captions.

3) The emulsion/suspention formation or bad solubility may highly influence MRI studies and relaxivity results. Please mention, in experimental and discussion sections, the compound status. Please, present the relaxation per concentration lines with R squared coefficient.

4) Moreover, the same thing may influence the cell viability. I see Figure 9 the cell viability higher 100 %. How it can be? Not good control? The data should be recalculated, and some explanations should be presented in the text. The nanoparticles may precipitate in solution and don't influence the cells. How you controlled it?

In Figure 15, what was used as 100% viability control? I see two 100% viability columns.

5) In section 3.2, I see the colloidal stability in deionized (DI) water. Please, present intensity mode. It may show colloidal stability in a better way.

6) However, have you studied the stability and aggregation in salt, at various pH, cell media, plasma (protein corona), etc.? It is an essential, organism media is not a deionized (DI) water. And the cells experiment is not the same as the presence in human plasma.

7) Have you studied the stability by TEM? If nanoparticles have bad colloidal stability, nanoparticles may precipitate in the DLS' cuvette and you may not see high-sized nanoparticles. In this way, DLS may not show anything.

8) It will be excellent to discuss before the conclusion how the size and shape is influenced on your results in one more small section as a Table or Figure with + or - marks or smth. like this. And what is more important, why it is influenced in this way.

Author Response

REVIEWER 1:

The author presented the paper "Effect of the size and shape of dendronized iron oxide nanoparticles bearing a targeting ligand on MRI, magnetic hyperthermia, and photothermia properties - from suspension to in vitro studies". It is an interesting work. However, below are some questions that can improve the quality of the article.

1) The reference list should be improved. Many more 2-3 years review papers should be cited in the Introduction section to show the progress in the area. I highly recommend not to use the references older 10 years for this section if it is possible.

Response: We agree that it would be better to add some more recent review papers in the introduction. New references have been added in the introduction part with many papers ranging from 2019 to 2023.

2) The first thing that is important for treatment and MRI is solubility in water. Please, clearly mention for the resulted system their colloidal stability, solubility in water. However, I have read done the text that they form an emulsion. It is not good for biomedical application.

Response: We have performed DLS measurement in order to assess the colloidal stability of our suspensions in water. The results are presented in Figure 4. The suspensions present a good colloidal stability in water with a monomodal size distribution.

We would like to mention that we cannot talk about solubility with nanoparticles as they are suspended in water. Finally, we are not sure to have clearly understood what you are meaning by speaking about “emulsion” because our suspensions do not form an emulsion and we did not speak about “emulsion” in our paper.

Please mention the solvent and concentration of nanoparticles and their form in this solution (good solubility, emulsion, etc.) for all methods in experimental sections and figure captions.

Response: The nanoparticles form a colloidal suspension in water and they do not dissolve or form emulsion. We speak about colloidal suspension and we have put “colloidal suspension” instead of “suspension” throughout the paper to be clearer.

We agree that we did not indicate the concentration in every figure. We corrected this both in the experimental section and in the figure captions.

3) The emulsion/suspension formation or bad solubility may highly influence MRI studies and relaxivity results. Please mention, in experimental and discussion sections, the compound status. Please, present the relaxation per concentration lines with R squared coefficient.

Response: The nanoparticles form a colloidal suspension in water, which is confirmed by DLS measurements in Figure 4 and the mean hydrodynamic size is reported in Table 3 for all samples including the commercial Resovist sample.

The relaxation time measurements have been detailed in the part 3.3 where we present our MRI results. We have added the relaxation per concentration with R squared coefficient in the supplementary information for one of the nanoparticle suspension as this measurement is representative of all performed measurements.

4) Moreover, the same thing may influence the cell viability. I see Figure 9 the cell viability higher 100 %. How it can be? Not good control? The data should be recalculated, and some explanations should be presented in the text. The nanoparticles may precipitate in solution and don't influence the cells. How you controlled it?

Response: The cell viability is higher than 100% because there were a higher number of cells in those wells than in the control wells. It is quite common for MTT tests to have results above 100% when the cells are not affected by the presence of the nanoparticles. As the number in the control well is fixed to 100% viability it is not surprising to have results above 100% (around 110 – 120%) for other wells.

The nanoparticles may indeed precipitate in solutions which could have an impact on cell viability as the cells are adherent to the wells. We made several washes with PBS to be sure to wash the nanoparticles from the cell layer.

5) In Figure 15, what was used as 100% viability control? I see two 100% viability columns.

Response: The 100% column corresponds to the cells in the control condition (no nanoparticles) and no laser irradiation. There is a second 100% because the number of cells found with the condition DNP-plates+no laser was equal to the control one. The results were normalized to this control condition (as for figure 9).

6) In section 3.2, I see the colloidal stability in deionized (DI) water. Please, present intensity mode. It may show colloidal stability in a better way.

Response: We have added the colloidal stability in intensity mode in the supplementary information. We had precised in the 2.2.8 part that “It must be noted that for shaped IONPs, this measurement is an indication of a possible aggregation but cannot be considered as a precise measurement of the hydrodynamic diameter as the optical model used to calculate the size distribution considered spherical NPs».

7) However, have you studied the stability and aggregation in salt, at various pH, cell media, plasma (protein corona), etc.? It is an essential, organism media is not a deionized (DI) water. And the cell experiment is not the same as the presence in human plasma.

Response: We completely agree with your remark, we have studied the colloidal stability in salt (NaCl) to mimic physiological media (Figure 11) and we see an aggregation phenomenon in salt. We have used these data to explain the higher cell internalization of nanocubes and nanoplates.

8) Have you studied the stability by TEM? If nanoparticles have bad colloidal stability, nanoparticles may precipitate in the DLS' cuvette and you may not see high-sized nanoparticles. In this way, DLS may not show anything.

Response: We don’t understand your remark as TEM does not allow the evaluation of the colloidal stability. When you deposit a drop of the nanoparticles suspension, an aggregation may occur during the drying step on the TEM grid. However, we can confirm that in water, we did not see precipitation phenomenon in the cuvettes for 24 hrs, but with salt, we could see that there were aggregates and that the suspension was more turbid with time in agreement with DLS measurements.

9) It will be excellent to discuss before the conclusion how the size and shape is influenced on your results in one more small section as a Table or Figure with + or - marks or smth. like this. And what is more important, why it is influenced in this way.

Response: We agree with your suggestion and have added a Table to conclude on the influence of size and shape on the results (Table 7).

Reviewer 2 Report

This is a very interesting and elaborate study on iron oxide nanoparticles based systems, having different shapes and being functionalized with targeting peptide, while being colloidally stabilized with dendron molecules. The IONPs can be utilized for MRI, magnetic hyperthermia, and photothermia applications as elegantly demonstrated. The authors characterized the nanosystems in detail using a gamut of techniques and explored their cytotoxicity and bioimaging/therapeutic properties in vitro. There are some minor points which should be further addressed.

1. A central question that comes to mind is, what are the reasons for obtaining IONPs of different shapes? Please add some details/clarifications on the formation process in each case.

2. Section 3.2: Can the extend of ligand exchange be quantified?

3. In relation to the previous comment: the chemical structures of the dendrons utilized should be presented and their design discussed in relation to IONPs functionalization.

4. Fig. 4:  please also present the intensity weighted size distributions.

5. Some typos should be corrected.

Author Response

REVIEWER 2:

This is a very interesting and elaborate study on iron oxide nanoparticles based systems, having different shapes and being functionalized with targeting peptide, while being colloidally stabilized with dendron molecules. The IONPs can be utilized for MRI, magnetic hyperthermia, and photothermia applications as elegantly demonstrated. The authors characterized the nanosystems in detail using a gamut of techniques and explored their cytotoxicity and bioimaging/therapeutic properties in vitro. There are some minor points which should be further addressed.

  1. A central question that comes to mind is, what are the reasons for obtaining IONPs of different shapes? Please add some details/clarifications on the formation process in each case.

Response: We have tested different iron oxide shaped nanoparticles to test the effect of an added shape anisotropy and saturation magnetization on MH. For PTT, we wanted to check if there is an effect of size and shape or not as, for the moment, the effect of size and shape on PTT is always in discussion. The introduction was slightly adapted to make this point clearer.

The formation process of each NP tested in this paper is deeply detailed in the material and method part.

  1. Section 3.2: Can the extent of ligand exchange be quantified?

Response: Yes, it can be quantified by elementary analysis of the phosphorus and iron with ICP-AES. We have highly explained the ligand exchange process between oleic acid and dendron molecules in previous papers, as well as the ICP procedure:

  1. Basly, G. Popa, S. Fleutot, B. P. Pichon, A. Garofalo, C. Ghobril, C. Billotey, A. Berniard, P. Bonazza, H. Martinez, D. Felder-Flesch and S. Begin-Colin, Dalton Trans., 2013, 42, 2146–2157.
  2. Bordeianu, A. Parat, C. Affolter-Zbaraszczuk, R. N. Muller, S. Boutry, S. Begin-Colin, F. Meyer, S. Laurent and D. Felder-Flesch, J. Mater. Chem. B, 2017, 5, 5152–5164.

References 13, 17, 33, 34 of the paper:

  1. Cotin, G.; Blanco-Andujar, C.; Nguyen, D.-V.; Affolter, C.; Boutry, S.; Boos, A.; Ronot, P.; Uring-Lambert, B.; Choquet, P.; Zorn, P.E.; et al. Dendron Based Antifouling, MRI and Magnetic Hyperthermia Properties of Different Shaped Iron Oxide Nanoparticles. Nanotechnology 2019, 30, 374002, doi:10.1088/1361-6528/ab2998.
  2. Cotin, G.; Blanco-Andujar, C.; Perton, F.; Asín, L.; Fuente, J.M. de la; Reichardt, W.; Schaffner, D.; Ngyen, D.-V.; Mertz, D.; Kiefer, C.; et al. Unveiling the Role of Surface, Size, Shape and Defects of Iron Oxide Nanoparticles for Theranostic Applications. Nanoscale 2021, 13, 14552–14571, doi:10.1039/D1NR03335B.
  3. Walter, A.; Garofalo, A.; Bonazza, P.; Meyer, F.; Martinez, H.; Fleutot, S.; Billotey, C.; Taleb, J.; Felder-Flesch, D.; Begin-Colin, S. Effect of the Functionalization Process on the Colloidal, Magnetic Resonance Imaging, and Bioelimination Properties of Mono- or Bisphosphonate-Anchored Dendronized Iron Oxide Nanoparticles. ChemPlusChem 2017, 82, 647–659, doi:10.1002/cplu.201700049.
  4. Walter, A.; Garofalo, A.; Parat, A.; Jouhannaud, J.; Pourroy, G.; Voirin, E.; Laurent, S.; Bonazza, P.; Taleb, J.; Billotey, C.; et al. Validation of a Dendron Concept to Tune Colloidal Stability, MRI Relaxivity and Bioelimination of Functional Nanoparticles. J. Mater. Chem. B 2015, 3, 1484–1494, doi:10.1039/C4TB01954G.

For 10 nm IONPs for example, we have approx. 700 dendron molecules per NP and a grafting rate of 1.4 dendron/nm2. The ICP-AES elementary analysis investigation was also performed for the various shaped IONPs and was added in the paper (Table 2).

  1. In relation to the previous comment: the chemical structures of the dendrons utilized should be presented and their design discussed in relation to IONPs functionalization.

Response: We agree that it is better to present the structure of the dendron. Its structure and its use was highly presented in our previous papers. So we have added the structure in supplementary information (Figure S3).

  1. Fig. 4: please also present the intensity weighted size distributions.

Response: We have added the intensity size distribution in the supplementary information. We had precised in the 2.2.8 part that “It must be noted that for shaped IONPs, this measurement is an indication of a possible aggregation but cannot be considered as a precise measurement of the hydrodynamic diameter as the optical model used to calculate the size distribution considered spherical NPs».

  1. Some typos should be corrected.

We have read again carefully the papers and have corrected typo mistakes.

Reviewer 3 Report

The novelty of this work needs to be elaborated more. It must be clearly emphasized added-value brought by this study to already existing specialty literature.

The current manuscript is more like a technical report rather than a research paper. The research contents are listed linearly, without highlighting the innovation and without convincing explanations regarding the majority of the statements issued.

For example:

(1)   at line 508:

 “The nanoplates present a higher Ms than previous reported nanoplates (16.7 ± 5.2 (length), 5.7 ± 1.6 (thickness) and Ms = 43 ± 5 emu/g) [17,43] against 20 – 25 nm length and 7 – 8 nm thickness for our nanoplates batch) due to their higher length and thickness.”

The authors should clearly explain the connection between Ms and the length/thickness of the nanoplates.

(2)   at line 517:

It is interesting to note that for the nanocubes, the size effect is quite visible when comparing MS values of the 19 nm and the 14 nm cubes.

Given the fact that all the positive effects observed by the authors in this paper have their origin in the high values obtained for Ms, the size effect on the saturation magnetigation is essential to be explained correctly.

The attempted explanation presented by the authors “The Ms increase could be explained by a larger fraction of magnetite as Baaziz et al. also experienced [9].”,  it is not sufficient.

(3)   at line 527:

The maximum of the ZFC/FC derivative curve can be assimilated to the IONPs blocking temperature (Figure 2.D). Values are reported on Table 2. This was also observed by Baaziz et al. [9] and Cotin et al.[17]. A shift toward higher temperatures is visible for the nanoacubes as they present the larger size.

The authors present only one result without trying to give any explanation for the increase in blocking temperature with the size of the nanocubes.

Author Response

REVIEWER 3:

The novelty of this work needs to be elaborated more. It must be clearly emphasized added-value brought by this study to already existing specialty literature.

Response: the introduction part has been modified to take into account your remarks.

The current manuscript is more like a technical report rather than a research paper. The research contents are listed linearly, without highlighting the innovation and without convincing explanations regarding the majority of the statements issued.

For example:

(1)   at line 508:

 “The nanoplates present a higher Ms than previous reported nanoplates (16.7 ± 5.2 (length), 5.7 ± 1.6 (thickness) and Ms = 43 ± 5 emu/g) [17,43] against 20 – 25 nm length and 7 – 8 nm thickness for our nanoplates batch) due to their higher length and thickness.”

The authors should clearly explain the connection between Ms and the length/thickness of the nanoplates.

Response: As both nanoplates are composed of oxidized magnetite (determined from XRD study), we can assess that there is more magnetite in the larger nanoplates and it explained also their higher Ms as the Ms of bulk magnetite is higher than that of bulk maghemite.

 (2)   at line 517:

“It is interesting to note that for the nanocubes, the size effect is quite visible when comparing Ms values of the 19 nm and the 14 nm cubes.”

Given the fact that all the positive effects observed by the authors in this paper have their origin in the high values obtained for Ms, the size effect on the saturation magnetization is essential to be explained correctly.

The attempted explanation presented by the authors “The Ms increase could be explained by a larger fraction of magnetite as Baaziz et al. also experienced [9].”,  it is not sufficient.

We have added this sentence to explain more the effect of size on Ms values: the saturation magnetization increases with the NPs size due to less surface effect like the spin canting effect and fewer defects. In addition, the higher is the size, the lower is the oxidation of nanoparticles because the oxidation occurs from the surface. We have demonstrated in previous publications that the coating with dendron prevent then nanoparticles from oxidation.

References:

  1. Baaziz, B. P. Pichon, S. Fleutot, Y. Liu, C. Lefevre, J.-M. Greneche, M. Toumi, T. Mhiri and S. Begin-Colin, J. Phys. Chem. C, 2014, 118, 3795–3810.
  2. Santoyo Salazar, L. Perez, O. de Abril, L. Truong Phuoc, D. Ihiawakrim, M. Vazquez, J.-M. Greneche, S. Begin-Colin and G. Pourroy, Chem. Mater., 2011, 23, 1379–1386.
  3. T. Thanh, Magnetic Nanoparticles: From Fabrication to Clinical Applications, CRC Press, Boca Raton, FL, 2012.

(3)   at line 527:

“The maximum of the ZFC/FC derivative curve can be assimilated to the IONPs blocking temperature (Figure 2.D). Values are reported on Table 2. This was also observed by Baaziz et al. [9] and Cotin et al.[17]. A shift toward higher temperatures is visible for the nanoacubes as they present the larger size.”

The authors present only one result without trying to give any explanation for the increase in blocking temperature with the size of the nanocubes.

Response: The authors agree that this part should be better discussed. This part has been completely rewritten: “The Verwey transition around 100 K, characteristic of the magnetite phase, is observed in ZFC/FC curves of NP_cubes_19, NP_cubes_14 and nanoplates. (highlighted in yellow in figure 2). It would confirm that Ms values may be mainly explained by the presence of a higher amount of magnetite. The maximum of the ZFC/FC derivative curve is often assimilated to the IONPs blocking temperature (Figure 2.D). The blocking temperature values reported in Table 2 depend on the characteristic of nanoparticules and also on their aggregation state. Values are in agreement with reported values for nanospheres [43]. The value for nanocubes may be related to their higher size: a shift toward higher temperatures is observed for the nanocubes with the highest size. The blocking temperature value of nanoplates is close to the one of spherical NP_12 but is difficult to discuss as this batch contains a large proportion of smaller spherical IONPs. »

Round 2

Reviewer 1 Report

Thank you for the revised version.

Reviewer 3 Report

I'm sorry that you failed to clearly emphasize the novelty and the added value brought by this study to the already existing specialized literature, because it would have certainly increased the citation rate of this otherwise interesting article.